# PipeGCN: Efficient Full-Graph Training of Graph Convolutional Networks with Pipelined Feature Communication

**Cheng Wan**
Rice University
chwan@rice.edu

**Youjie Li**
UIUC
li238@illinois.edu

**Cameron R. Wolfe**
Rice University
crw13@rice.edu

**Anastasios Kyrillidis**
Rice University
anastasios@rice.edu

**Nam Sung Kim**
UIUC
nam.sung.kim@gmail.com

**Yingyan Lin**
Rice University
yl150@rice.edu

## Abstract

Graph Convolutional Networks (GCNs) is the state-of-the-art method for learning graph-structured data, and training large-scale GCNs requires distributed training across multiple accelerators such that each accelerator is able to hold a partitioned subgraph. However, distributed GCN training incurs prohibitive overhead of communicating node features and feature gradients among partitions for every GCN layer during each training iteration, limiting the achievable training efficiency and model scalability. To this end, we propose PipeGCN, a simple yet effective scheme that hides the communication overhead by pipelining inter-partition communication with intra-partition computation. It is non-trivial to pipeline for efficient GCN training, as communicated node features/gradients will become stale and thus can harm the convergence, negating the pipeline benefit. Notably, little is known regarding the convergence rate of GCN training with both stale features and stale feature gradients. This work not only provides a theoretical convergence analysis but also finds the convergence rate of PipeGCN to be close to that of the vanilla distributed GCN training without any staleness. Furthermore, we develop a smoothing method to further improve PipeGCN's convergence. Extensive experiments show that PipeGCN can largely boost the training throughput ($1.7\times\sim28.5\times$) while achieving the same accuracy as its vanilla counterpart and existing full-graph training methods. The code is available at https://github.com/RICE-EIC/PipeGCN.

## 1 Introduction

Graph Convolutional Networks (GCNs) (Kipf & Welling, 2016) have gained great popularity recently as they demonstrated the state-of-the-art (SOTA) performance in learning graph-structured data (Zhang & Chen, 2018; Xu et al., 2018; Ying et al., 2018). Their promising performance is resulting from their ability to capture diverse neighborhood connectivity. In particular, a GCN aggregates all features from the neighbor node set for a given node, the feature of which is then updated via a multi-layer perceptron. Such a two-step process (*neighbor aggregation* and *node update*) empowers GCNs to better learn graph structures. Despite their promising performance, training GCNs at scale is still a challenging problem, as a prohibitive amount of compute and memory resources are required to train a real-world large-scale graph, let alone exploring deeper and more advanced models. To overcome this challenge, various sampling-based methods have been proposed to reduce the resource requirement at a cost of incurring feature approximation errors. A straightforward instance is to create mini-batches by sampling neighbors (e.g., GraphSAGE (Hamilton et al., 2017) and VR-GCN (Chen et al., 2018)) or to extract subgraphs as training samples (e.g., Cluster-GCN (Chiang et al., 2019) and GraphSAINT (Zeng et al., 2020)).

In addition to sampling-based methods, distributed GCN training has emerged as a promising alternative, as it enables large *full-graph* training of GCNs across multiple accelerators such as GPUs.

This approach first partitions a giant graph into multiple small subgraps, each of which is able to fit into a single GPU, and then train these partitioned subgraphs locally on GPUs together with indispensable communication across partitions. Following this direction, several recent works (Ma et al., 2019; Jia et al., 2020; Tripathy et al., 2020; Thorpe et al., 2021; Wan et al., 2022) have been proposed and verified the great potential of distributed GCN training. $P^3$ (Gandhi & Iyer, 2021) follows another direction that splits the data along the feature dimension and leverages intra-layer model parallelism for training, which shows superior performance on small models.

In this work, we propose a new method for distributed GCN training, PipeGCN, which targets achieving a full-graph accuracy with boosted training efficiency. Our main contributions are following:

- We first analyze two efficiency bottlenecks in distributed GCN training: the required *significant communication overhead* and *frequent synchronization*, and then propose a simple yet effective technique called PipeGCN to address both aforementioned bottlenecks by pipelining inter-partition communication with intra-partition computation to hide the communication overhead.

- We address the challenge raised by PipeGCN, i.e., the resulting staleness in communicated features and feature gradients (*neither weights nor weight gradients*), by providing a theoretical convergence analysis and showing that PipeGCN's convergence rate is $\mathcal{O}(T^{-\frac{2}{3}})$, i.e., close to vanilla distributed GCN training without staleness. *To the best of our knowledge, we are the first to provide a theoretical convergence proof of GCN training with both stale feature and stale feature gradients.*

- We further propose a low-overhead smoothing method to further improve PipeGCN's convergence by reducing the error incurred by the staleness.

- Extensive empirical and ablation studies consistently validate the advantages of PipeGCN over both vanilla distributed GCN training and those SOTA full-graph training methods (e.g., *boosting the training throughput by 1.7×∼28.5× while achieving the same or a better accuracy*).

## 2 BACKGROUND AND RELATED WORKS

**Graph Convolutional Networks.** GCNs represent each node in a graph as a feature (embedding) vector and learn the feature vector via a two-step process (*neighbor aggregation* and then *node update*) for each layer, which can be mathematically described as:

$$z_v^{(\ell)} = \zeta^{(\ell)} \left( \left\{ h_u^{(\ell-1)} \mid u \in \mathcal{N}(v) \right\} \right) \tag{1}$$

$$h_v^{(\ell)} = \phi^{(\ell)} \left( z_v^{(\ell)}, h_v^{(\ell-1)} \right) \tag{2}$$

where $\mathcal{N}(v)$ is the neighbor set of node $v$ in the graph, $h_v^{(\ell)}$ represents the learned embedding vector of node $v$ at the $\ell$-th layer, $z_v^{(\ell)}$ is an intermediate aggregated feature calculated by an aggregation function $\zeta^{(\ell)}$, and $\phi^{(\ell)}$ is the function for updating the feature of node $v$. The original GCN (Kipf & Welling, 2016) uses a weighted average aggregator for $\zeta^{(\ell)}$ and the update function $\phi^{(\ell)}$ is a single-layer perceptron $\sigma(W^{(\ell)} z_v^{(\ell)})$ where $\sigma(\cdot)$ is a non-linear activation function and $W^{(\ell)}$ is a weight matrix. Another famous GCN instance is GraphSAGE (Hamilton et al., 2017) in which $\phi^{(\ell)}$ is $\sigma \left( W^{(\ell)} \cdot \text{CONCAT} \left( z_v^{(\ell)}, h_v^{(\ell-1)} \right) \right)$.

**Distributed Training for GCNs.** A real-world graph can contain millions of nodes and billions of edges (Hu et al., 2020), for which a feasible training approach is to partition it into small subgraphs (to fit into each GPU's resource), and train them in parallel, during which necessary communication is performed to exchange boundary node features and gradients to satisfy GCNs's *neighbor aggregation* (Equ. 1). Such an approach is called *vanilla partition-parallel training* and is illustrated in Fig. 1 (a). Following this approach, several works have been proposed recently. NeuGraph (Ma et al., 2019), AliGraph (Zhu et al., 2019), and ROC (Jia et al., 2020) perform such partition-parallel training but rely on CPUs for storage for all partitions and repeated swapping of a partial partition to GPUs. Inevitably, prohibitive CPU-GPU swaps are incurred, plaguing the achievable training efficiency. CAGNET (Tripathy et al., 2020) is different in that it splits each node feature vector into tiny sub-vectors which are then broadcasted and computed sequentially, thus requiring redundant communication and frequent synchronization. Furthermore, $P^3$ (Gandhi & Iyer, 2021) proposes to split both the feature and the GCN layer for mitigating the communication overhead, but it makes a strong assumption that the hidden dimensions of a GCN should be considerably smaller than that of

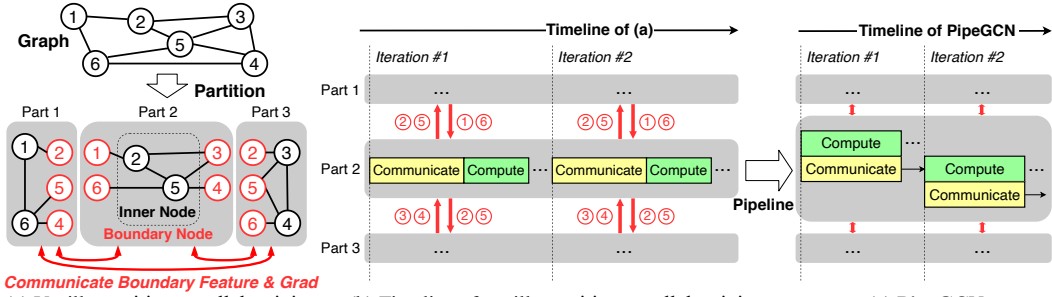

Figure 1: An illustrative comparison between vanilla partition-parallel training and PipeGCN.

input features, which restricts the model size. A concurrent work Dorylus (Thorpe et al., 2021) adopts a fine-grained pipeline along each compute operation in GCN training and supports asynchronous usage of stale features. Nevertheless, the resulting staleness of *feature gradients* is neither analyzed nor considered for convergence proof, let alone error reduction methods for the incurred staleness.

**Asynchronous Distributed Training.** Many prior works have been proposed for asynchronous distributed training of DNNs. Most works (e.g., Hogwild! (Niu et al., 2011), SSP (Ho et al., 2013), and MXNet (Li et al., 2014)) rely on a parameter server with multiple workers running asynchronously to hide communication overhead of *weights/(weight gradients)* among each other, at a cost of using stale *weight gradients* from previous iterations. Other works like Pipe-SGD (Li et al.,

Table 1: Differences between conventional asynchronous distributed training and PipeGCN.

| Method | Hogwild!, SSP, MXNet, Pipe-SGD, PipeDream, PipeMare | **PipeGCN** |
|---|---|---|
| Target | Large Model, Small Feature | Large Feature |
| Staleness | Weight Gradients | Features and Feature Gradients |

2018b) pipeline such communication with local computation of each worker. Another direction is to partition a large model along its layers across multiple GPUs and then stream in small data batches through the layer pipeline, e.g., PipeDream (Harlap et al., 2018) and PipeMare (Yang et al., 2021). Nonetheless, all these works aim at large models with small data, where communication overhead of model *weights/weight gradients* are substantial but data feature communications are marginal (if not none), thus not well suited for GCNs. More importantly, they focus on convergence with stale *weight gradients* of models, rather than stale *features/feature gradients* incurred in GCN training. Tab. 1 summarizes the differences. In a nutshell, *little effort has been made to study asynchronous or pipelined distributed training of GCNs, where feature communication plays the major role, let alone the corresponding theoretical convergence proofs.*

**GCNs with Stale Features/Feature Gradients.** Several recent works have been proposed to adopt either stale features (Chen et al., 2018; Cong et al., 2020) or feature gradients (Cong et al., 2021) in single-GPU training of GCNs. Nevertheless, their convergence analysis considers only one of two kinds of staleness and derives a convergence rate of $\mathcal{O}(T^{-\frac{1}{2}})$ for pure sampling-based methods. This is, however, limited in distributed GCN training as its *convergence is simultaneously affected by both kinds of staleness*. PipeGCN proves such convergence with both stale features and feature gradients and offers a better rate of $\mathcal{O}(T^{-\frac{2}{3}})$. Furthermore, none of previous works has studied the errors incurred by staleness which harms the convergence speed, while PipeGCN develops a low-overhead smoothing method to reduce such errors.

## 3 THE PROPOSED PIPEGCN FRAMEWORK

**Overview.** To enable efficient distributed GCN training, we first identify the two bottlenecks associated with vanilla partition-parallel training: *substantial communication overhead* and *frequently synchronized communication* (see Fig. 1(b)), and then address them directly by proposing a novel strategy, PipeGCN, which pipelines the communication and computation stages across two adjacent iterations in each partition of distributed GCN training for breaking the synchrony and then hiding the communication latency (see Fig. 1(c)). It is non-trivial to achieve efficient GCN training with such a pipeline method, as staleness is incurred in communicated features/feature gradients and

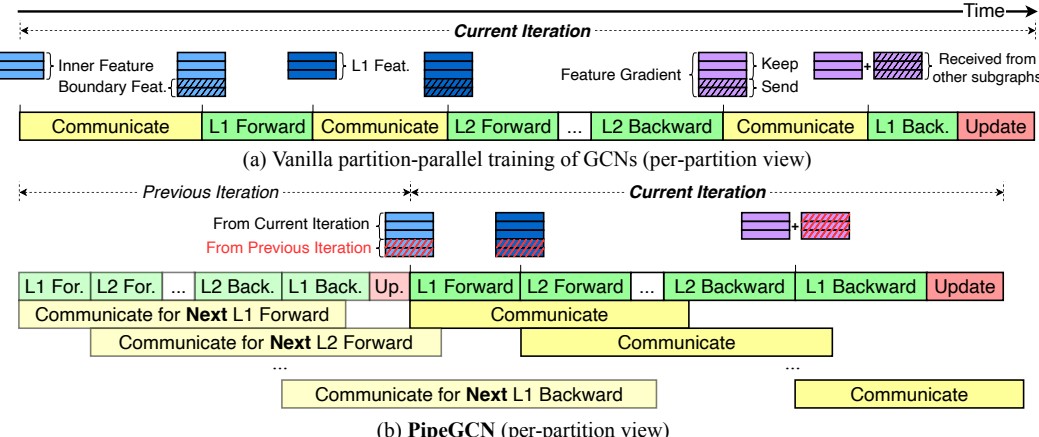

Figure 2: A detailed comparison between vanilla partition-parallel training of GCNs and PipeGCN.

more importantly little effort has been made to study the convergence guarantee of GCN training using stale feature gradients. This work takes an initial effort to prove both the theoretical and empirical convergence of such a pipelined GCN training method, and for the first time shows its convergence rate to be close to that of vanilla GCN training without staleness. Furthermore, we propose a low-overhead smoothing method to reduce the errors due to stale features/feature gradients for further improving the convergence.

## 3.1 BOTTLENECKS IN VANILLA PARTITION-PARALLEL TRAINING

**Significant communication overhead.** Fig. 1(a) illustrates *vanilla partition-parallel training*, where each partition holds *inner nodes* that come from the original graph and *boundary nodes* that come from other subgraphs. These boundary nodes are demanded by the *neighbor aggregation* of GCNs across neighbor partitions, e.g., in Fig. 1(a) node-5 needs nodes-[3,4,6] from other partitions for calculating Equ. 1. Therefore, *it is the features/gradients of boundary nodes that dominate the communication overhead in distributed GCN training*. Note that the amount of boundary nodes can be excessive and far exceeds the inner nodes, as the

Table 2: The substantial communication overhead in vanilla partition-parallel training, where *Comm. Ratio* is the communication time divided by the total training time.

| Dataset | # Partition | Comm. Ratio |
|---|---|---|
| Reddit | 2 | 65.83% |
| | 4 | 82.89% |
| ogbn-products | 5 | 76.17% |
| | 10 | 85.79% |
| Yelp | 3 | 61.16% |
| | 6 | 76.84% |

boundary nodes are replicated across partitions and scale with the number of partitions. Besides the sheer size, communication of boundary nodes occurs for (1) each layer and (2) both forward and backward passes, making communication overhead substantial. We evaluate such overhead[1] in Tab. 2 and find communication to be dominant, which is consistent with CAGNET (Tripathy et al., 2020).

**Frequently synchronized communication.** The aforementioned communication of boundary nodes must be finished before calculating Equ. 1 and Equ. 2, which inevitably *forces synchronization between communication and computation and requires a fully sequential execution* (see Fig. 1(b)). Thus, for most of training time, each partition is waiting for dominant features/gradients communication to finish before the actual compute, repeated for each layer and for both forward and backward passes.

## 3.2 THE PROPOSED PIPEGCN METHOD

Fig. 1(c) illustrates the high-level overview of PipeGCN, which pipelines the *communicate* and *compute* stages spanning two iterations for each GCN layer. Fig. 2 further provides the detailed end-to-end flow, where PipeGCN removes the heavy communication overhead in the vanilla approach by breaking the synchronization between communicate and compute and hiding communicate with compute of each GCN layer. This is achieved by *deferring the communicate to next iteration's compute* (instead of serving the current iteration) such that compute and communicate can run in

---

[1]The detailed setting can be found in Sec. 4.

---

**Algorithm 1:** Training a GCN with PipeGCN (per-partition view).

---

**Input:** partition id $i$, partition count $n$, graph partition $\mathcal{G}_i$, propagation matrix $P_i$, node feature
$\quad\quad X_i$, label $Y_i$, boundary node set $\mathcal{B}_i$, layer count $L$, learning rate $\eta$, initial model $W_0$
**Output:** trained model $W_T$ after $T$ iterations

1   $\mathcal{V}_i \leftarrow \{\text{node } v \in \mathcal{G}_i : v \notin \mathcal{B}_i\}$            $\triangleright$ create inner node set
2   Broadcast $\mathcal{B}_i$ and Receive $[\mathcal{B}_1, \cdots, \mathcal{B}_n]$
3   $[\mathcal{S}_{i,1}, \cdots, \mathcal{S}_{i,n}] \leftarrow [\mathcal{B}_1 \cap \mathcal{V}_i, \cdots, \mathcal{B}_n \cap \mathcal{V}_i]$
4   Broadcast $\mathcal{V}_i$ and Receive $[\mathcal{V}_1, \cdots, \mathcal{V}_n]$
5   $[\mathcal{S}_{1,i}, \cdots, \mathcal{S}_{n,i}] \leftarrow [\mathcal{B}_i \cap \mathcal{V}_1, \cdots, \mathcal{B}_i \cap \mathcal{V}_n]$
6   $H^{(0)} \leftarrow \begin{bmatrix} X_i \\ 0 \end{bmatrix}$           $\triangleright$ initialize node feature, set boundary feature as 0
7   **for** $t := 1 \rightarrow T$ **do**
8     **for** $\ell := 1 \rightarrow L$ **do**           $\triangleright$ forward pass
9       **if** $t > 1$ **then**
10         wait until $thread_f^{(\ell)}$ completes
11         $[H_{\mathcal{S}_{1,i}}^{(\ell-1)}, \cdots, H_{\mathcal{S}_{n,i}}^{(\ell-1)}] \leftarrow [B_1^{(\ell)}, \cdots, B_n^{(\ell)}]$      $\triangleright$ update boundary feature
12       **end**
13       **with** $thread_f^{(\ell)}$         $\triangleright$ communicate boundary features in parallel
14         Send $[H_{\mathcal{S}_{i,1}}^{(\ell-1)}, \cdots, H_{\mathcal{S}_{i,n}}^{(\ell-1)}]$ to partition $[1, \cdots, n]$ and Receive $[B_1^{(\ell)}, \cdots, B_n^{(\ell)}]$
15       $H_{\mathcal{V}_i}^{(\ell)} \leftarrow \sigma(P_i H^{(\ell-1)} W_{t-1}^{(\ell)})$       $\triangleright$ update inner nodes feature
16     **end**
17     $J_{\mathcal{V}_i}^{(L)} \leftarrow \dfrac{\partial Loss(H_{\mathcal{V}_i}^{(L)}, Y_i)}{\partial H_{\mathcal{V}_i}^{(L)}}$
18     **for** $\ell := L \rightarrow 1$ **do**           $\triangleright$ backward pass
19       $G_i^{(\ell)} \leftarrow \left[P_i H^{(\ell-1)}\right]^{\top} \left(J_{\mathcal{V}_i}^{(\ell)} \circ \sigma'(P_i H^{(\ell-1)} W_{t-1}^{(\ell)})\right)$      $\triangleright$ calculate weight gradient
20       **if** $\ell > 1$ **then**
21         $J^{(\ell-1)} \leftarrow P_i^{\top} \left(J_{\mathcal{V}_i}^{(\ell)} \circ \sigma'(P_i H^{(\ell-1)} W_{t-1}^{(\ell)})\right) [W_{t-1}^{(\ell)}]^{\top}$      $\triangleright$ calculate feature gradient
22         **if** $t > 1$ **then**
23           wait until $thread_b^{(\ell)}$ completes
24           **for** $j := 1 \rightarrow n$ **do**
25             $J_{\mathcal{S}_{i,j}}^{(\ell-1)} \leftarrow J_{\mathcal{S}_{i,j}}^{(\ell-1)} + C_j^{(\ell)}$      $\triangleright$ accumulate feature gradient
26           **end**
27         **end**
28         **with** $thread_b^{(\ell)}$        $\triangleright$ communicate boundary feature gradient in parallel
29           Send $[J_{\mathcal{S}_{1,i}}^{(\ell-1)}, \cdots, J_{\mathcal{S}_{n,i}}^{(\ell-1)}]$ to partition $[1, \cdots, n]$ and Receive $[C_1^{(\ell)}, \cdots, C_n^{(\ell)}]$
30       **end**
31     **end**
32     $G \leftarrow AllReduce(G_i)$           $\triangleright$ synchronize model gradient
33     $W_t \leftarrow W_{t-1} - \eta \cdot G$           $\triangleright$ update model
34   **end**
35   **return** $W_T$

---

parallel. Inevitably, staleness is introduced in the deferred communication and results in *a mixture usage of fresh inner features/gradients and staled boundary features/gradients*.

Analytically, PipeGCN is achieved by modifying Equ. 1. For instance, when using a mean aggregator, Equ. 1 and its corresponding backward formulation in PipeGCN become:

$$z_v^{(t,\ell)} = \text{MEAN}\left(\{h_u^{(t,\ell-1)} \mid u \in \mathcal{N}(v) \setminus \mathcal{B}(v)\} \cup \{h_u^{(t-1,\ell-1)} \mid u \in \mathcal{B}(v)\}\right) \tag{3}$$

$$\delta_{h_u}^{(t,\ell)} = \sum_{v:u\in\mathcal{N}(v)\setminus\mathcal{B}(v)} \frac{1}{d_v} \cdot \delta_{z_v}^{(t,\ell+1)} + \sum_{v:u\in\mathcal{B}(v)} \frac{1}{d_v} \cdot \delta_{z_v}^{(t-1,\ell+1)} \tag{4}$$

where $\mathcal{B}(v)$ is node $v$'s boundary node set, $d_v$ denotes node $v$'s degree, and $\delta_{h_u}^{(t,\ell)}$ and $\delta_{z_v}^{(t,\ell)}$ represent the gradient approximation of $h_u$ and $z_v$ at layer $\ell$ and iteration $t$, respectively. Lastly, the implementation of PipeGCN are outlined in Alg. 1.

### 3.3 PipeGCN's Convergence Guarantee

As PipeGCN adopts a mixture usage of fresh inner features/gradients and staled boundary features/gradients, its convergence rate is still unknown. We have proved the convergence of PipeGCN and present the convergence property in the following theorem.

**Theorem 3.1** (Convergence of PipeGCN, informal version). *There exists a constant $E$ such that for any arbitrarily small constant $\varepsilon > 0$, we can choose a learning rate $\eta = \frac{\sqrt{\varepsilon}}{E}$ and number of training iterations $T = (\mathcal{L}(\theta^{(1)}) - \mathcal{L}(\theta^*))E\varepsilon^{-\frac{3}{2}}$ such that:*

$$\frac{1}{T}\sum_{t=1}^{T}\|\nabla\mathcal{L}(\theta^{(t)})\|_2 \leq \mathcal{O}(\varepsilon)$$

*where $\mathcal{L}(\cdot)$ is the loss function, $\theta^{(t)}$ and $\theta^*$ represent the parameter vector at iteration $t$ and the optimal parameter respectively.*

Therefore, **the convergence rate of PipeGCN is $\mathcal{O}(T^{-\frac{2}{3}})$, which is better than sampling-based method ($\mathcal{O}(T^{-\frac{1}{2}})$)** (Chen et al., 2018; Cong et al., 2021) **and close to full-graph training ($\mathcal{O}(T^{-1})$).** The formal version of the theorem and our detailed proof can be founded in Appendix A.

### 3.4 The Proposed Smoothing Method

To further improve the convergence of PipeGCN, we propose a smoothing method to reduce errors incurred by stale features/feature gradients at a minimal overhead. Here we present the smoothing of feature gradients, and the same formulation also applies to features. To improve the approximate gradients for each feature, fluctuations in feature gradients between adjacent iterations should be reduced. Therefore, we apply a light-weight moving average to the feature gradients of each boundary node $v$ as follow:

$$\hat{\delta}_{z_v}^{(t,\ell)} = \gamma\hat{\delta}_{z_v}^{(t-1,\ell)} + (1-\gamma)\delta_{z_v}^{(t,\ell)}$$

where $\hat{\delta}_{z_v}^{(t,\ell)}$ is the smoothed feature gradient at layer $\ell$ and iteration $t$, and $\gamma$ is the decay rate. When integrating this smoothed feature gradient method into the backward pass, Equ. 4 can be rewritten as:

$$\hat{\delta}_{h_u}^{(t,\ell)} = \sum_{v:u\in\mathcal{N}(v)\setminus\mathcal{B}(v)}\frac{1}{d_v}\cdot\delta_{z_v}^{(t,\ell+1)} + \sum_{v:u\in\mathcal{B}(v)}\frac{1}{d_v}\cdot\hat{\delta}_{z_v}^{(t-1,\ell+1)}$$

Note that the smoothing of stale features and gradients can be independently applied to PipeGCN.

## 4 Experiment Results

We evaluate PipeGCN on four large-scale datasets, Reddit (Hamilton et al., 2017), ogbn-products (Hu et al., 2020), Yelp (Zeng et al., 2020), and ogbn-papers100M (Hu et al., 2020). More details are provided in Tab. 3. ***To ensure robustness and reproducibility, we fix (i.e., do not tune) the hyperparameters and settings for PipeGCN and its variants throughout all experiments***. To implement partition parallelism (for both vanilla distributed GCN training and PipeGCN), the widely used METIS (Karypis & Kumar, 1998) partition algorithm is adopted for graph partition with its objective set to minimize the communication volume. We implement PipeGCN in PyTorch (Paszke et al., 2019) and DGL (Wang et al., 2019). Experiments are conducted on a machine with 10 RTX-2080Ti (11GB), Xeon 6230R@2.10GHz (187GB), and PCIe3x16 connecting CPU-GPU and GPU-GPU. Only for ogbn-papers100M, we use 4 compute nodes (each contains 8 MI60 GPUs, an AMD EPYC 7642 CPU, and 48 lane PCI 3.0 connecting CPU-GPU and GPU-GPU) networked with 10Gbps Ethernet. To support full-graph GCN training with the model sizes in Tab. 3, the minimum required partition numbers are 2, 3, 5, 32 for Reddit, ogbn-products, Yelp, and ogbn-papers100M, respectively.

For convenience, we here name all methods: vanilla partition-parallel training of GCNs (**GCN**), PipeGCN with feature gradient smoothing (**PipeGCN-G**), PipeGCN with feature smoothing (**PipeGCN-F**), and PipeGCN with both smoothing (**PipeGCN-GF**). The default decay rate $\gamma$ for all smoothing methods is set to 0.95.

Table 3: Detailed experiment setups: graph datasets, GCN models, and training hyper-parameters.

| Dataset | # Nodes | # Edges | Feat. size | GraphSAGE model size | Optimizer | LearnRate | Dropout | # Epoch |
|---|---|---|---|---|---|---|---|---|
| Reddit | 233K | 114M | 602 | 4 layer, 256 hidden units | Adam | 0.01 | 0.5 | 3000 |
| ogbn-products | 2.4M | 62M | 100 | 3 layer, 128 hidden units | Adam | 0.003 | 0.3 | 500 |
| Yelp | 716K | 7.0M | 300 | 4 layer, 512 hidden units | Adam | 0.001 | 0.1 | 3000 |
| ogbn-papers100M | 111M | 1.6B | 128 | 3 layer, 48 hidden units | Adam | 0.01 | 0.0 | 1000 |

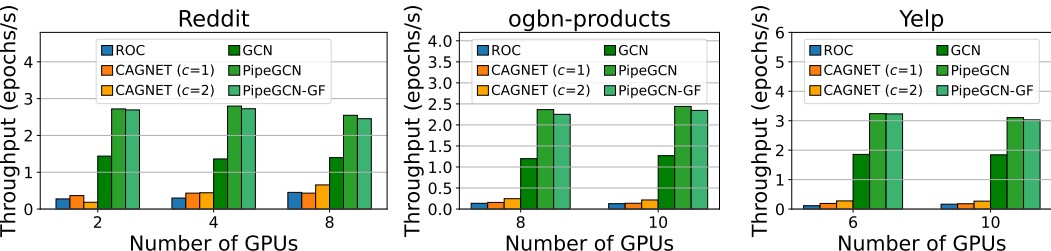

Figure 3: Throughput comparison. Each partition uses one GPU (except CAGNET ($c$=2) uses two).

## 4.1 IMPROVING TRAINING THROUGHPUT OVER FULL-GRAPH TRAINING METHODS

Fig. 3 compares the training throughput between PipeGCN and the SOTA full-graph training methods (ROC (Jia et al., 2020) and CAGNET (Tripathy et al., 2020)). We observe that both vanilla partition-parallel training (GCN) and PipeGCN greatly outperform ROC and CAGNET across different number of partitions, because they avoid both the expensive CPU-GPU swaps (ROC) and the redundant node broadcast (CAGNET). Specifically, GCN is **3.1×∼16.4×** faster than ROC and **2.1×∼10.2×** faster than CAGNET ($c$=2). PipeGCN further improves upon GCN, achieving a throughput improvement of **5.6×∼28.5×** over ROC and **3.9×∼17.7×** over CAGNET ($c$=2)[2]. Note that we are not able to compare PipeGCN with NeuGraph (Ma et al., 2019), AliGraph (Zhu et al., 2019), and $P^3$ (Gandhi & Iyer, 2021) as their code are not publicly available. Besides, Dorylus (Thorpe et al., 2021) is not comparable, as it is not for regular GPU servers. *Considering the substantial performance gap between ROC/CAGNET and GCN, we focus on comparing GCN with PipeGCN for the reminder of the section.*

## 4.2 IMPROVING TRAINING THROUGHPUT WITHOUT COMPROMISING ACCURACY

We compare the training performance of both test score and training throughput between GCN and PipeGCN in Tab. 4. We can see that *PipeGCN without smoothing already achieves a comparable test score with the vanilla GCN training* on both Reddit and Yelp, and incurs only a negligible accuracy drop (-0.08%∼-0.23%) on ogbn-products, while boosting the training throughput by **1.72×∼2.16×** across all datasets and different number of partitions[3], thus validating the effectiveness of PipeGCN.

With the proposed smoothing method plugged in, *PipeGCN-G/F/GF is able to compensate the dropped score of vanilla PipeGCN, achieving an equal or even better test score as/than the vanilla GCN training* (without staleness), e.g., 97.14% vs. 97.11% on Reddit, 79.36% vs. 79.14% on ogbn-products and 65.28% vs. 65.26% on Yelp. Meanwhile, PipeGCN-G/F/GF enjoys a similar throughput improvement as vanilla PipeGCN, thus validating the negligible overhead of the proposed smoothing method. Therefore, ***pipelined transfer of features and gradients greatly improves the training throughput while maintaining the full-graph accuracy***.

Note that our distributed GCN training methods consistently achieve higher test scores than SOTA sampling-based methods for GraphSAGE-based models reported in (Zeng et al., 2020) and (Hu et al., 2020), confirming that full-graph training is preferred to obtain better GCN models. For example, the best sampling-based method achieves a 96.6% accuracy on Reddit (Zeng et al., 2020) while full-graph GCN training achieves 97.1%, and PipeGCN improves the accuracy by 0.28% over sampling-based GraphSAGE models on ogbn-products (Hu et al., 2020). This advantage of full-graph training is also validated by recent works (Jia et al., 2020; Tripathy et al., 2020; Liu et al., 2022; Wan et al., 2022).

---

[2]More detailed comparisons among full-graph training methods can be found in Appendix B.

[3]More details regarding PipeGCN's advantages in training throughput can be found in Appendix C.

Table 4: Training performance comparison among vanilla partition-parallel training (GCN) and PipeGCN variants (PipeGCN*), where we report the test accuracy for Reddit and ogbn-products, and the F1-micro score for Yelp. Highest performance is in bold.

| Dataset | Method | Test Score (%) | Throughput | Dataset | Method | Test Score (%) | Throughput |
|---------|--------|----------------|------------|---------|--------|----------------|------------|
| Reddit (2 partitions) | GCN | 97.11±0.02 | 1× (1.94 epochs/s) | Reddit (4 partitions) | GCN | **97.11±0.02** | 1× (2.07 epochs/s) |
| | PipeGCN | 97.12±0.02 | **1.91×** | | PipeGCN | 97.04±0.03 | **2.12×** |
| | PipeGCN-G | **97.14±0.03** | 1.89× | | PipeGCN-G | 97.09±0.03 | 2.07× |
| | PipeGCN-F | 97.09±0.02 | 1.89× | | PipeGCN-F | 97.10±0.02 | 2.10× |
| | PipeGCN-GF | 97.12±0.02 | 1.87× | | PipeGCN-GF | 97.10±0.02 | 2.06× |
| ogbn-products (5 partitions) | GCN | 79.14±0.35 | 1× (1.45 epochs/s) | ogbn-products (10 partitions) | GCN | 79.14±0.35 | 1× (1.28 epochs/s) |
| | PipeGCN | 79.06±0.42 | **1.94×** | | PipeGCN | 78.91±0.65 | **1.87×** |
| | PipeGCN-G | 79.20±0.38 | 1.90× | | PipeGCN-G | 79.08±0.58 | 1.82× |
| | PipeGCN-F | **79.36±0.38** | 1.90× | | PipeGCN-F | **79.21±0.31** | 1.81× |
| | PipeGCN-GF | 78.86±0.34 | 1.91× | | PipeGCN-GF | 78.77±0.23 | 1.82× |
| Yelp (3 partitions) | GCN | 65.26±0.02 | 1× (2.00 epochs/s) | Yelp (6 partitions) | GCN | 65.26±0.02 | 1× (2.25 epochs/s) |
| | PipeGCN | **65.27±0.01** | **2.16×** | | PipeGCN | 65.24±0.02 | **1.72×** |
| | PipeGCN-G | 65.26±0.02 | 2.15× | | PipeGCN-G | **65.28±0.02** | 1.69× |
| | PipeGCN-F | 65.26±0.03 | 2.15× | | PipeGCN-F | 65.25±0.04 | 1.68× |
| | PipeGCN-GF | 65.26±0.04 | 2.11× | | PipeGCN-GF | 65.26±0.04 | 1.67× |

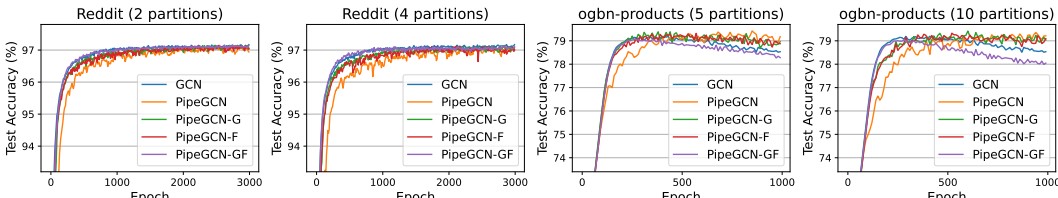

Figure 4: Epoch-to-accuracy comparison among vanilla partition-parallel training (GCN) and PipeGCN variants (PipeGCN*), where *PipeGCN and its variants achieve a similar convergence as the vanilla training (without staleness) but are twice as fast in wall-clock time* (see Tab. 4).

### 4.3 MAINTAINING CONVERGENCE SPEED

To understand PipeGCN's influence on the convergence speed, we compare the training curve among different methods in Fig. 4. We observe that the convergence of PipeGCN without smoothing is still comparable with that of the vanilla GCN training, although PipeGCN converges slower at the early phase of training and then catches up at the later phase, due to the staleness of boundary features/gradients. With the proposed smoothing methods, *PipeGCN-G/F boosts the convergence substantially and matches the convergence speed of vanilla GCN training*. There is no clear difference between PipeGCN-G and PipeGCN-F. Lastly, with combined smoothing of features and gradients, *PipeGCN-GF can acheive the same or even slightly better convergence speed as vanilla GCN training* (e.g., on Reddit) but can overfit gradually similar to the vanilla GCN training, which is further investigated in Sec. 4.4. Therefore, ***PipeGCN maintains the convergence speed w.r.t the number of epochs while reduces the end-to-end training time by around 50% thanks to its boosted training throughput*** (see Tab. 4).

### 4.4 BENEFIT OF STALENESS SMOOTHING METHOD

**Error Reduction and Convergence Speedup.** To understand why the proposed smoothing technique (Sec. 3.4) speeds up convergence, we compare the error incurred by the stale communication between PipeGCN and PipeGCN-G/F. The error is calculated as the Frobenius-norm of the gap between the correct gradient/feature and the stale gradient/feature used in PipeGCN training. Fig. 5 compares the error at each GCN layer. We can see that *the proposed smoothing technique (PipeGCN-G/F) reduces the error of staleness substantially* (from the base version of PipeGCN) and this benefit consistently holds across different layers in terms of both feature and gradient errors, validating the effectiveness of our smoothing method and explaining its improvement to the convergence speed.

**Overfitting Mitigation.** To understand the effect of staleness smoothing on model overfitting, we also evaluate the test-accuracy convergence under different decay rates $\gamma$ in Fig. 6. Here ogbn-products is adopted as the study case because the distribution of its test set largely differs from that of its training set. From Fig. 6, we observe that smoothing with a large $\gamma$ (0.7/0.95) offers a fast convergence, i.e., close to the vanilla GCN training, but overfits rapidly. To understand this issue, we

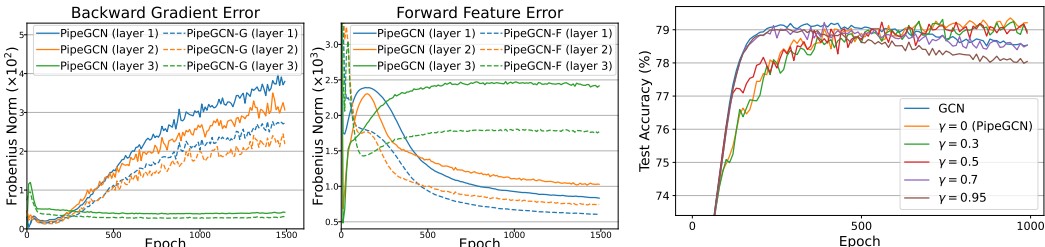

Figure 5: Comparison of the resulting feature gradient error and feature error from PipeGCN and PipeGCN-G/F at each GCN layer on Reddit (2 partitions). PipeGCN-G/F here uses a default smoothing decay rate of 0.95.

Figure 6: Test-accuracy convergence comparison among different smoothing decay rates $\gamma$ in PipeGCN-GF on ogbn-products (10 partitions).

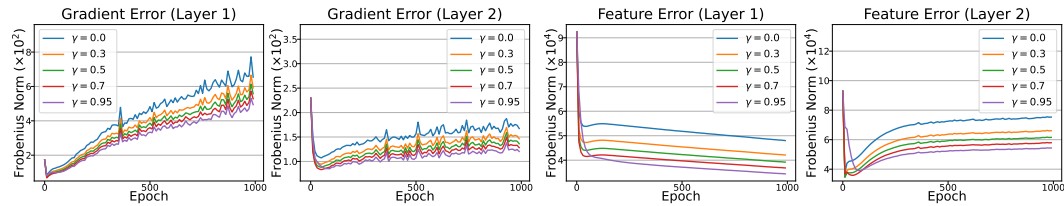

Figure 7: Comparison of the resulting feature gradient error and feature error when adopting different decay rates $\gamma$ at each GCN layer on ogbn-products (10 partitions).

further provide detailed comparisons of the errors incurred under different $\gamma$ in Fig. 7. We can see that a larger $\gamma$ enjoys lower approximation errors and makes the gradients/features more stable, thus improving the convergence speed. The increased stability on the training set, however, constrains the model from exploring a more general minimum point on the test set, thus leading to overfitting as the vanilla GCN training. In contrast, *a small $\gamma$ ($0 \sim 0.5$) mitigates this overfitting and achieves a better accuracy* (see Fig. 6). But a too-small $\gamma$ (e.g., 0) gives a high error for both stale features and gradients (see Fig. 7), thus suffering from a slower convergence. Therefore, a trade-off between convergence speed and achievable optimality exists between different smoothing decay rates, and $\gamma = 0.5$ combines the best of both worlds in this study.

## 4.5 SCALING LARGE GRAPH TRAINING OVER MULTIPLE SERVERS

To further test the capability of PipeGCN, we scale up the graph size to ogbn-papers100M and train GCN over multiple GPU servers with 32 GPUs. Tab. 5 shows that even at such a large-scale setting where communication overhead dominates, PipeGCN still reduce communication time by 61%, leading to a total training time reduction of 38% compared to the vanilla GCN baseline [4].

Table 5: Comparison of epoch training time on ogbn-papers100M.

| Method | Total | Communication |
|---|---|---|
| GCN | $1.00\times$ (10.5s) | $1.00\times$ (6.6s) |
| PipeGCN | $0.62\times$ (6.5s) | $0.39\times$ (2.6s) |
| PipeGCN-GF | $0.64\times$ (6.7s) | $0.42\times$ (2.8s) |

## 5 CONCLUSION

In this work, we propose a new method, PipeGCN, for efficient full-graph GCN training. PipeGCN pipelines communication with computation in distributed GCN training to hide the prohibitive communication overhead. More importantly, we are the first to provide convergence analysis for GCN training with both stale features and feature gradients, and further propose a light-weight smoothing method for convergence speedup. Extensive experiments validate the advantages of PipeGCN over both vanilla GCN training (without staleness) and state-of-the-art full-graph training.

---

[4]More experiments on multi-server training can be found in Appendix E.

## 6 ACKNOWLEDGEMENT

The work is supported by the National Science Foundation (NSF) through the MLWiNS program (Award number: 2003137), the CC* Compute program (Award number: 2019007), and the NeTS program (Award number: 1801865).

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

# A  CONVERGENCE PROOF

In this section, we prove the convergence of PipeGCN. Specifically, we first figure out that when the model is updated via gradient descent, the change of intermediate features and their gradients are bounded by a constant which is proportional to learning rate $\eta$ under standard assumptions. Based on this, we further demonstrate that the error occurred by the staleness is proportional to $\eta$, which guarantees that the gradient error is bounded by $\eta E$ where $E$ is defined in Corollary A.10, and thus PipeGCN converges in $\mathcal{O}(\varepsilon^{-\frac{3}{2}})$ iterations.

## A.1  NOTATIONS AND ASSUMPTIONS

For a given graph $\mathcal{G} = (\mathcal{V}, \mathcal{E})$ with an adjacency matrix $A$, feature matrix $X$, we define the propagation matrix $P$ as $P := \widetilde{D}^{-1/2}\widetilde{A}\widetilde{D}^{-1/2}$, where $\widetilde{A} = A + I$, $\widetilde{D}_{u,u} = \sum_v \widetilde{A}_{u,v}$. One GCN layer performs one step of feature propagation (Kipf & Welling, 2016) as formulated below

$$H^{(0)} = X$$
$$Z^{(\ell)} = PH^{(\ell-1)}W^{(\ell)}$$
$$H^{(\ell)} = \sigma(Z^{(\ell)})$$

where $H^{(\ell)}$, $W^{(\ell)}$, and $Z^{(\ell)}$ denote the embedding matrix, the trainable weight matrix, and the intermediate embedding matrix in the $\ell$-th layer, respectively, and $\sigma$ denotes a non-linear activation function. For an $L$-layer GCN, the loss function is denoted by $\mathcal{L}(\theta)$ where $\theta = \text{vec}[W^{(1)}, W^{(2)}, \cdots, W^{(L)}]$. We define the $\ell$-th layer as a function $f^{(\ell)}(\cdot, \cdot)$.

$$f^{(\ell)}(H^{(\ell-1)}, W^{(\ell)}) := \sigma(PH^{(\ell-1)}W^{(\ell)})$$

Its gradient w.r.t. the input embedding matrix can be represented as

$$J^{(\ell-1)} = \nabla_H f^{(\ell)}(J^{(\ell)}, H^{(\ell-1)}, W^{(\ell)}) := P^\top M^{(\ell)}[W^{(\ell)}]^\top$$

and its gradient w.r.t. the weight can be represented as

$$G^{(\ell)} = \nabla_W f^{(\ell)}(J^{(\ell)}, H^{(\ell-1)}, W^{(\ell)}) := [PH^{(\ell-1)}]^\top M^{(\ell)}$$

where $M^{(\ell)} = J^{(\ell)} \circ \sigma'(PH^{(\ell-1)}W^{(\ell)})$ and $\circ$ denotes Hadamard product.

For partition-parallel training, we can split $P$ into two parts $P = P_{in} + P_{bd}$ where $P_{in}$ represents intra-partition propagation and $P_{bd}$ denotes inter-partition propagation. For PipeGCN, we can represent one GCN layer as below

$$\widetilde{H}^{(t,0)} = X$$
$$\widetilde{Z}^{(t,\ell)} = P_{in}\widetilde{H}^{(t,\ell-1)}\widetilde{W}^{(t,\ell)} + P_{bd}\widetilde{H}^{(t-1,\ell-1)}\widetilde{W}^{(t,\ell)}$$
$$\widetilde{H}^{(t,\ell)} = \sigma(\widetilde{Z}^{(t,\ell)})$$

where $t$ is the epoch number and $\widetilde{W}^{(t,\ell)}$ is the weight at epoch $t$ layer $\ell$. We define the loss function for this setting as $\widetilde{\mathcal{L}}(\widetilde{\theta}^{(t)})$ where $\widetilde{\theta}^{(t)} = \text{vec}[\widetilde{W}^{(t,1)}, \widetilde{W}^{(t,2)}, \cdots, \widetilde{W}^{(t,L)}]$. We can also summarize the layer as a function $\widetilde{f}^{(t,\ell)}(\cdot, \cdot)$

$$\widetilde{f}^{(t,\ell)}(\widetilde{H}^{(t,\ell-1)}, \widetilde{W}^{(t,\ell)}) := \sigma(P_{in}\widetilde{H}^{(t,\ell-1)}\widetilde{W}^{(t,\ell)} + P_{bd}\widetilde{H}^{(t-1,\ell-1)}\widetilde{W}^{(t,\ell)})$$

Note that $\widetilde{H}^{(t-1,\ell-1)}$ is not a part of the input of $\widetilde{f}^{(t,\ell)}(\cdot, \cdot)$ because it is a constant for the $t$-th epoch. The corresponding backward propagation follows the following computation

$$\widetilde{J}^{(t,\ell-1)} = \nabla_H \widetilde{f}^{(t,\ell)}(\widetilde{J}^{(t,\ell)}, \widetilde{H}^{(t,\ell-1)}, \widetilde{W}^{(t,\ell)})$$
$$\widetilde{G}^{(t,\ell)} = \nabla_W \widetilde{f}^{(t,\ell)}(\widetilde{J}^{(t,\ell)}, \widetilde{H}^{(t,\ell-1)}, \widetilde{W}^{(t,\ell)})$$

where

$$\widetilde{M}^{(t,\ell)} = \widetilde{J}^{(t,\ell)} \circ \sigma'(P_{in}\widetilde{H}^{(t,\ell-1)}\widetilde{W}^{(t,\ell)} + P_{bd}\widetilde{H}^{(t-1,\ell-1)}\widetilde{W}^{(t,\ell)})$$

$$\nabla_H \widetilde{f}^{(t,\ell)}(\widetilde{J}^{(t,\ell)}, \widetilde{H}^{(t,\ell-1)}, \widetilde{W}^{(t,\ell)}) := P_{in}^\top \widetilde{M}^{(t,\ell)}[\widetilde{W}^{(t,\ell)}]^\top + P_{bd}^\top \widetilde{M}^{(t-1,\ell)}[\widetilde{W}^{(t-1,\ell)}]^\top$$

$$\nabla_W \widetilde{f}^{(t,\ell)}(\widetilde{J}^{(t,\ell)}, \widetilde{H}^{(t,\ell-1)}, \widetilde{W}^{(t,\ell)}) := [P_{in}\widetilde{H}^{(t,\ell-1)} + P_{bd}\widetilde{H}^{(t-1,\ell-1)}]^\top \widetilde{M}^{(t,\ell)}$$

Again, $\widetilde{J}^{(t-1,\ell)}$ is not a part of the input of $\nabla_H \widetilde{f}^{(t,\ell)}(\cdot,\cdot,\cdot)$ or $\nabla_W \widetilde{f}^{(t,\ell)}(\cdot,\cdot,\cdot)$ because it is a constant for epoch $t$. Finally, we define $\nabla\widetilde{\mathcal{L}}(\widetilde{\theta}^{(t)}) = \text{vec}[\widetilde{G}^{(t,1)}, \widetilde{G}^{(t,2)}, \cdots, \widetilde{G}^{(t,L)}]$. It should be highlighted that the 'gradient' $\nabla_H \widetilde{f}^{(t,\ell)}(\cdot,\cdot,\cdot)$, $\nabla_W \widetilde{f}^{(t,\ell)}(\cdot,\cdot,\cdot)$ and $\nabla\widetilde{\mathcal{L}}(\widetilde{\theta}^{(t)})$ are not the standard gradient for the corresponding forward process due to the stale communication. Properties of gradient cannot be directly applied to these variables.

Before proceeding our proof, we make the following standard assumptions about the adopted GCN architecture and input graph.

**Assumption A.1.** *The loss function $Loss(\cdot, \cdot)$ is $C_{loss}$-Lipschitz continuous and $L_{loss}$-smooth w.r.t. to the input node embedding vector, i.e., $|Loss(h^{(L)}, y) - Loss(h'^{(L)}, y)| \leq C_{loss}\|h^{(L)} - h'^{(L)}\|_2$ and $\|\nabla Loss(h^{(L)}, y) - \nabla Loss(h'^{(L)}, y)\|_2 \leq L_{loss}\|h^{(L)} - h'^{(L)}\|_2$ where $h$ is the predicted label and $y$ is the correct label vector.*

**Assumption A.2.** *The activation function $\sigma(\cdot)$ is $C_\sigma$-Lipschitz continuous and $L_\sigma$-smooth, i.e., $\|\sigma(z^{(\ell)}) - \sigma(z'^{(\ell)})\|_2 \leq C_\sigma\|z^{(\ell)} - z'^{(\ell)}\|_2$ and $\|\sigma'(z^{(\ell)}) - \sigma'(z'^{(\ell)})\|_2 \leq L_\sigma\|z^{(\ell)} - z'^{(\ell)}\|_2$.*

**Assumption A.3.** *For any $\ell \in [L]$, the norm of weight matrices, the propagation matrix, and the input feature matrix are bounded: $\|W^{(\ell)}\|_F \leq B_W, \|P\|_F \leq B_P, \|X\|_F \leq B_X$. (This generic assumption is also used in (Chen et al., 2018; Liao et al., 2020; Garg et al., 2020; Cong et al., 2021).)*

### A.2 BOUNDED MATRICES AND CHANGES

**Lemma A.1.** *For any $\ell \in [L]$, the Frobenius norm of node embedding matrices, gradient passing from the $\ell$-th layer node embeddings to the $(\ell-1)$-th, gradient matrices are bounded, i.e.,*

$$\|H^{(\ell)}\|_F, \|\widetilde{H}^{(t,\ell)}\|_F \leq B_H,$$

$$\|J^{(\ell)}\|_F, \|\widetilde{J}^{(t,\ell)}\|_F \leq B_J,$$

$$\|M^{(\ell)}\|_F, \|\widetilde{M}^{(t,\ell)}\|_F \leq B_M,$$

$$\|G^{(\ell)}\|_F, \|\widetilde{G}^{(t,\ell)}\|_F \leq B_G$$

*where*

$$B_H = \max_{1 \leq \ell \leq L}(C_\sigma B_P B_W)^\ell B_X$$

$$B_J = \max_{2 \leq \ell \leq L}(C_\sigma B_P B_W)^{L-\ell}C_{loss}$$

$$B_M = C_\sigma B_J$$

$$B_G = B_P B_H B_M$$

*Proof.* The proof of $\|H^{(\ell)}\|_F \leq B_H$ and $\|J^{(\ell)}\|_F \leq B_J$ can be found in Proposition 1 in (Cong et al., 2021). By induction,

$$\|\widetilde{H}^{(t,\ell)}\|_F = \|\sigma(P_{in}\widetilde{H}^{(t,\ell-1)}\widetilde{W}^{(t,\ell)} + P_{bd}\widetilde{H}^{(t-1,\ell-1)}\widetilde{W}^{(t,\ell)})\|_F$$
$$\leq C_\sigma B_W\|P_{in} + P_{bd}\|_F(C_\sigma B_P B_W)^{\ell-1}B_X$$
$$\leq (C_\sigma B_P B_W)^\ell B_X$$

$$\|\widetilde{J}^{(t,\ell-1)}\|_F = \left\|P_{in}^\top\left(\widetilde{J}^{(t,\ell)} \circ \sigma'(\widetilde{Z}^{(t,\ell)})\right)[\widetilde{W}^{(t,\ell)}]^\top + P_{bd}^\top\left(\widetilde{J}^{(t-1,\ell)} \circ \sigma'(\widetilde{Z}^{(t-1,\ell)})\right)[\widetilde{W}^{(t-1,\ell)}]^\top\right\|_F$$
$$\leq C_\sigma B_W\|P_{in} + P_{bd}\|_F(C_\sigma B_P B_W)^{L-\ell}C_{loss}$$
$$\leq (C_\sigma B_P B_W)^{L-\ell+1}C_{loss}$$

$$\|M^{(\ell)}\|_F = \|J^{(\ell)} \circ \sigma'(Z^{(\ell)})\|_F \leq C_\sigma B_J$$
$$\|\widetilde{M}^{(t,\ell)}\|_F = \|\widetilde{J}^{(t,\ell)} \circ \sigma'(\widetilde{Z}^{(t,\ell)})\|_F \leq C_\sigma B_J$$

$$G^{(\ell)} = [PH^{(\ell-1)}]^\top M^{(\ell)}$$
$$\leq B_P B_H B_M$$

$$\widetilde{G}^{(t,\ell)} = [P_{in}\widetilde{H}^{(t,\ell-1)} + P_{bd}\widetilde{H}^{(t-1,\ell-1)}]^\top \widetilde{M}^{(t,\ell)}$$
$$\leq B_P B_H B_M$$

$\square$

Because the gradient matrices are bounded, the weight change is bounded.

**Corollary A.2.** *For any $t, \ell$, $\|\widetilde{W}^{(t,\ell)} - \widetilde{W}^{(t-1,\ell)}\|_F \leq B_{\Delta W} = \eta B_G$ where $\eta$ is the learning rate.*

Now we can analyze the changes of intermediate variables.

**Lemma A.3.** *For any $t, \ell$, we have $\|\widetilde{Z}^{(t,\ell)} - \widetilde{Z}^{(t-1,\ell)}\|_F \leq B_{\Delta Z}$, $\|\widetilde{H}^{(t,\ell)} - \widetilde{H}^{(t-1,\ell)}\|_F \leq B_{\Delta H}$, where $B_{\Delta Z} = \sum_{i=0}^{L-1} C_\sigma^i B_P^{i+1} B_W^i B_H B_{\Delta W}$ and $B_{\Delta H} = C_\sigma B_{\Delta Z}$.*

*Proof.* When $\ell = 0$, $\|\widetilde{H}^{(t,0)} - \widetilde{H}^{(t-1,0)}\|_F = \|X - X\|_F = 0$. Now we consider $\ell > 0$ by induction.

$$\|\widetilde{Z}^{(t,\ell)} - \widetilde{Z}^{(t-1,\ell)}\|_F = \|(P_{in}\widetilde{H}^{(t,\ell-1)}\widetilde{W}^{(t,\ell)} + P_{bd}\widetilde{H}^{(t-1,\ell-1)}\widetilde{W}^{(t,\ell)})$$
$$- (P_{in}\widetilde{H}^{(t-1,\ell-1)}\widetilde{W}^{(t-1,\ell)} + P_{bd}\widetilde{H}^{(t-2,\ell-1)}\widetilde{W}^{(t-1,\ell)})\|_F$$
$$= \|P_{in}(\widetilde{H}^{(t,\ell-1)}\widetilde{W}^{(t,\ell)} - \widetilde{H}^{(t-1,\ell-1)}\widetilde{W}^{(t-1,\ell)})$$
$$+ P_{bd}(\widetilde{H}^{(t-1,\ell-1)}\widetilde{W}^{(t,\ell)} - \widetilde{H}^{(t-2,\ell-1)}\widetilde{W}^{(t-1,\ell)})\|_F$$

Then we analyze the bound of $\|\widetilde{H}^{(t,\ell-1)}\widetilde{W}^{(t,\ell)} - \widetilde{H}^{(t-1,\ell-1)}\widetilde{W}^{(t-1,\ell)}\|_F$ which is denoted by $s^{(t,\ell)}$.

$$s^{(t,\ell)} \leq \|\widetilde{H}^{(t,\ell-1)}\widetilde{W}^{(t,\ell)} - \widetilde{H}^{(t,\ell-1)}\widetilde{W}^{(t-1,\ell)}\|_F + \|\widetilde{H}^{(t,\ell-1)}\widetilde{W}^{(t-1,\ell)} - \widetilde{H}^{(t-1,\ell-1)}\widetilde{W}^{(t-1,\ell)}\|_F$$
$$\leq B_H\|\widetilde{W}^{(t,\ell)} - \widetilde{W}^{(t-1,\ell)}\|_F + B_W\|\widetilde{H}^{(t,\ell-1)} - \widetilde{H}^{(t-1,\ell-1)}\|_F$$

According to Corollary A.2, $\|\widetilde{W}^{(t,\ell)} - \widetilde{W}^{(t-1,\ell)}\|_F \leq B_{\Delta W}$. By induction, $\|\widetilde{H}^{(t,\ell-1)} - \widetilde{H}^{(t-1,\ell-1)}\|_F \leq \sum_{i=0}^{\ell-2} C_\sigma^{i+1} B_P^{i+1} B_W^i B_H B_{\Delta W}$. Combining these inequalities,

$$s^{(t,\ell)} \leq B_H B_{\Delta W} + \sum_{i=1}^{\ell-1} C_\sigma^i B_P^i B_W^i B_H B_{\Delta W}$$

Plugging it back, we have

$$\|\widetilde{Z}^{(t,\ell)} - \widetilde{Z}^{(t-1,\ell)}\|_F \leq \|P_{in}(\widetilde{H}^{(t,\ell-1)}\widetilde{W}^{(t,\ell)} - \widetilde{H}^{(t-1,\ell-1)}\widetilde{W}^{(t-1,\ell)})$$
$$+ P_{bd}(\widetilde{H}^{(t-1,\ell-1)}\widetilde{W}^{(t,\ell)} - \widetilde{H}^{(t-2,\ell-1)}\widetilde{W}^{(t-1,\ell)})\|_F$$
$$\leq B_P \left( B_H B_{\Delta W} + \sum_{i=1}^{\ell-1} C_\sigma^i B_P^i B_W^i B_H B_{\Delta W} \right)$$
$$= \sum_{i=0}^{\ell-1} C_\sigma^i B_P^{i+1} B_W^i B_H B_{\Delta W}$$

$$\|\widetilde{H}^{(t,\ell)} - \widetilde{H}^{(t-1,\ell)}\|_F = \|\sigma(\widetilde{Z}^{(t,\ell)}) - \sigma(\widetilde{Z}^{(t-1,\ell)})\|_F$$
$$\leq C_\sigma\|\widetilde{Z}^{(t,\ell)} - \widetilde{Z}^{(t-1,\ell)}\|_F$$
$$\leq C_\sigma B_{\Delta Z}$$

$\square$

**Lemma A.4.** $\|\widetilde{J}^{(t,\ell)} - \widetilde{J}^{(t-1,\ell)}\|_F \leq B_{\Delta J}$ *where*

$$B_{\Delta J} = \max_{2 \leq \ell \leq L} (B_P B_W C_\sigma)^{L-\ell} B_{\Delta H} L_{loss} + (B_M B_{\Delta W} + L_\sigma B_J B_{\Delta Z} B_W) \sum_{i=0}^{L-3} B_P^{i+1} B_W^i C_\sigma^i$$

*Proof.* For the last layer ($\ell = L$), $\|\widetilde{J}^{(t,L)} - \widetilde{J}^{(t-1,L)}\|_F \leq L_{\text{loss}}\|\widetilde{H}^{(t,L)} - \widetilde{H}^{(t-1,L)}\|_F \leq L_{\text{loss}} B_{\Delta H}$. For the case of $\ell < L$, we prove the lemma by using induction.

$$\begin{aligned}
\|\widetilde{J}^{(t,\ell-1)} - \widetilde{J}^{(t-1,\ell-1)}\|_F &= \left\| \left( P_{in}^\top \widetilde{M}^{(t,\ell)} [\widetilde{W}^{(t,\ell)}]^\top + P_{bd}^\top \widetilde{M}^{(t-1,\ell)} [\widetilde{W}^{(t-1,\ell)}]^\top \right) \right. \\
&\quad \left. - \left( P_{in}^\top \widetilde{M}^{(t-1,\ell)} [\widetilde{W}^{(t-1,\ell)}]^\top + P_{bd}^\top \widetilde{M}^{(t-2,\ell)} [\widetilde{W}^{(t-2,\ell)}]^\top \right) \right\|_F \\
&\leq \left\| P_{in}^\top \left( \widetilde{M}^{(t,\ell)} [\widetilde{W}^{(t,\ell)}]^\top - \widetilde{M}^{(t-1,\ell)} [\widetilde{W}^{(t-1,\ell)}]^\top \right) \right\|_F \\
&\quad + \left\| P_{bd}^\top \left( \widetilde{M}^{(t-1,\ell)} [\widetilde{W}^{(t-1,\ell)}]^\top - \widetilde{M}^{(t-2,\ell)} [\widetilde{W}^{(t-2,\ell)}]^\top \right) \right\|_F
\end{aligned}$$

We denote $\left\| \widetilde{M}^{(t,\ell)} [\widetilde{W}^{(t,\ell)}]^\top - \widetilde{M}^{(t-1,\ell)} [\widetilde{W}^{(t-1,\ell)}]^\top \right\|_F$ by $s^{(t,\ell)}$ and analyze its bound.

$$\begin{aligned}
s^{(t,\ell)} &\leq \left\| \widetilde{M}^{(t,\ell)} [\widetilde{W}^{(t,\ell)}]^\top - \widetilde{M}^{(t,\ell)} [\widetilde{W}^{(t-1,\ell)}]^\top \right\|_F \\
&\quad + \left\| \widetilde{M}^{(t,\ell)} [\widetilde{W}^{(t-1,\ell)}]^\top - \widetilde{M}^{(t-1,\ell)} [\widetilde{W}^{(t-1,\ell)}]^\top \right\|_F \\
&\leq B_M \left\| [\widetilde{W}^{(t,\ell)}]^\top - [\widetilde{W}^{(t-1,\ell)}]^\top \right\|_F + B_W \left\| \widetilde{M}^{(t,\ell)} - \widetilde{M}^{(t-1,\ell)} \right\|_F
\end{aligned}$$

According to Corollary A.2, $\left\| [\widetilde{W}^{(t,\ell)}]^\top - [\widetilde{W}^{(t-1,\ell)}]^\top \right\|_F \leq B_{\Delta W}$. For the second term,

$$\begin{aligned}
&\|\widetilde{M}^{(t,\ell)} - \widetilde{M}^{(t-1,\ell)}\|_F \\
=&\|\widetilde{J}^{(t,\ell)} \circ \sigma'(\widetilde{Z}^{(t,\ell)}) - \widetilde{J}^{(t-1,\ell)} \circ \sigma'(\widetilde{Z}^{(t-1,\ell)})\|_F \\
\leq&\|\widetilde{J}^{(t,\ell)} \circ \sigma'(\widetilde{Z}^{(t,\ell)}) - \widetilde{J}^{(t,\ell)} \circ \sigma'(\widetilde{Z}^{(t-1,\ell)})\|_F + \|\widetilde{J}^{(t,\ell)} \circ \sigma'(\widetilde{Z}^{(t-1,\ell)}) - \widetilde{J}^{(t-1,\ell)} \circ \sigma'(\widetilde{Z}^{(t-1,\ell)})\|_F \\
\leq&B_J\|\sigma'(\widetilde{Z}^{(t,\ell)}) - \sigma'(\widetilde{Z}^{(t-1,\ell)})\|_F + C_\sigma\|\widetilde{J}^{(t,\ell)} - \widetilde{J}^{(t-1,\ell)}\|_F \qquad (5)
\end{aligned}$$

According to the smoothness of $\sigma$ and Lemma A.3, $\|\sigma'(\widetilde{Z}^{(t,\ell)}) - \sigma'(\widetilde{Z}^{(t-1,\ell)})\|_F \leq L_\sigma B_{\Delta Z}$. By induction,

$$\begin{aligned}
&\|\widetilde{J}^{(t,\ell)} - \widetilde{J}^{(t-1,\ell)}\|_F \\
&\leq (B_P B_W C_\sigma)^{(L-\ell)} B_{\Delta H} L_{\text{loss}} + (B_M B_{\Delta W} + L_\sigma B_J B_{\Delta Z} B_W) \sum_{i=0}^{L-\ell-1} B_P^{i+1} B_W^i C_\sigma^i
\end{aligned}$$

As a result,

$$\begin{aligned}
s^{(t,\ell)} &\leq B_M B_{\Delta W} + B_W B_J L_\sigma B_{\Delta Z} + B_W C_\sigma \|\widetilde{J}^{(t,\ell)} - \widetilde{J}^{(t-1,\ell)}\|_F \\
&= (B_M B_{\Delta W} + B_W B_J L_\sigma B_{\Delta Z}) + B_P^{(L-\ell)} B_W^{(L-\ell+1)} C_\sigma^{(L-\ell+1)} B_{\Delta H} L_{\text{loss}} \\
&\quad + (B_M B_{\Delta W} + L_\sigma B_J B_{\Delta Z} B_W) \sum_{i=1}^{L-\ell} B_P^i B_W^i C_\sigma^i \\
&\leq B_P^{(L-\ell)} B_W^{(L-\ell+1)} C_\sigma^{(L-\ell+1)} B_{\Delta H} L_{\text{loss}} \\
&\quad + (B_M B_{\Delta W} + L_\sigma B_J B_{\Delta Z} B_W) \sum_{i=0}^{L-\ell} B_P^i B_W^i C_\sigma^i
\end{aligned}$$

$$\|\widetilde{J}^{(t,\ell-1)} - \widetilde{J}^{(t-1,\ell-1)}\|_F = \left\|P_{in}^\top \left(\widetilde{M}^{(t,\ell)}[\widetilde{W}^{(t,\ell)}]^\top - \widetilde{M}^{(t-1,\ell)}[\widetilde{W}^{(t-1,\ell)}]^\top\right)\right\|_F$$

$$+ \left\|P_{bd}^\top \left(\widetilde{M}^{(t-1,\ell)}[\widetilde{W}^{(t-1,\ell)}]^\top - \widetilde{M}^{(t-2,\ell)}[\widetilde{W}^{(t-2,\ell)}]^\top\right)\right\|_F$$

$$\leq B_P s^{(t,\ell)}$$

$$\leq (B_P B_W C_\sigma)^{(L-\ell+1)} B_{\Delta H} L_{\text{loss}}$$

$$+ (B_M B_{\Delta W} + L_\sigma B_J B_{\Delta Z} B_W) \sum_{i=0}^{L-\ell} B_P^{i+1} B_W^i C_\sigma^i$$

$\square$

From Equation 5, we can also conclude that

**Corollary A.5.** $\|\widetilde{M}^{(t,\ell)} - \widetilde{M}^{(t-1,\ell)}\|_F \leq B_{\Delta M}$ with $B_{\Delta M} = B_J L_\sigma B_{\Delta Z} + C_\sigma B_{\Delta J}$.

### A.3 BOUNDED FEATURE ERROR AND GRADIENT ERROR

In this subsection, we compare the difference between generic GCN and PipeGCN with the same parameter set, i.e., $\theta = \widetilde{\theta}^{(t)}$.

**Lemma A.6.** $\|\widetilde{Z}^{(t,\ell)} - Z^{(\ell)}\|_F \leq E_Z, \|\widetilde{H}^{(t,\ell)} - H^{(\ell)}\|_F \leq E_H$ where $E_Z = B_{\Delta H} \sum_{i=1}^{L} C_\sigma^{i-1} B_W^i B_P^i$

and $E_H = B_{\Delta H} \sum_{i=1}^{L} (C_\sigma B_W B_P)^i$.

*Proof.*

$$\|\widetilde{Z}^{(t,\ell)} - Z^{(\ell)}\|_F = \|(P_{in}\widetilde{H}^{(t,\ell-1)}\widetilde{W}^{(t,\ell)} + P_{bd}\widetilde{H}^{(t-1,\ell-1)}\widetilde{W}^{(t,\ell)}) - (PH^{(\ell-1)}W^{(\ell)})\|_F$$

$$\leq \|(P_{in}\widetilde{H}^{(t,\ell-1)} + P_{bd}\widetilde{H}^{(t-1,\ell-1)} - PH^{(\ell-1)})W^{(\ell)}\|_F$$

$$= B_W \|P(\widetilde{H}^{(t,\ell-1)} - H^{(\ell-1)}) + P_{bd}(\widetilde{H}^{(t-1,\ell-1)} - \widetilde{H}^{(t,\ell-1)})\|_F$$

$$\leq B_W B_P \left(\|\widetilde{H}^{(t,\ell-1)} - H^{(\ell-1)}\|_F + B_{\Delta H}\right)$$

By induction, we assume that $\|\widetilde{H}^{(t,\ell-1)} - H^{(\ell-1)}\|_F \leq B_{\Delta H} \sum_{i=1}^{\ell-1} (C_\sigma B_W B_P)^i$. Therefore,

$$\|\widetilde{Z}^{(t,\ell)} - Z^{(\ell)}\|_F \leq B_W B_P B_{\Delta H} \sum_{i=0}^{\ell-1} (C_\sigma B_W B_P)^i$$

$$= B_{\Delta H} \sum_{i=1}^{\ell} C_\sigma^{i-1} B_W^i B_P^i$$

$$\|\widetilde{H}^{(t,\ell)} - H^{(\ell)}\|_F = \|\sigma(\widetilde{Z}^{(t,\ell)}) - \sigma(Z^{(\ell)})\|_F$$

$$\leq C_\sigma \|\widetilde{Z}^{(t,\ell)} - Z^{(\ell)}\|_F$$

$$\leq B_{\Delta H} \sum_{i=1}^{\ell} (C_\sigma B_W B_P)^i$$

$\square$

**Lemma A.7.** $\|\widetilde{J}^{(t,\ell)} - J^{(\ell)}\|_F \leq E_J$ and $\|\widetilde{M}^{(t,\ell)} - M^{(\ell)}\|_F \leq E_M$ with

$$E_J = \max_{2 \leq \ell \leq L} (B_P B_W C_\sigma)^{L-\ell} L_{loss} E_H + B_P (B_W (B_J E_Z L_\sigma + B_{\Delta M}) + B_{\Delta W} B_M) \sum_{i=0}^{L-3} (B_P B_W C_\sigma)^i$$

$$E_M = C_\sigma E_J + L_\sigma B_J E_Z$$

*Proof.* When $\ell = L$, $\|\widetilde{J}^{(t,L)} - J^{(L)}\|_F \leq L_{\text{loss}}E_H$. For any $\ell$, we assume that

$$\|\widetilde{J}^{(t,\ell)} - J^{(\ell)}\|_F \leq (B_P B_W C_\sigma)^{L-\ell} L_{\text{loss}} E_H + U \sum_{i=0}^{L-\ell-1} (B_P B_W C_\sigma)^i \tag{6}$$

$$\|\widetilde{M}^{(t,\ell)} - M^{(\ell)}\|_F \leq (B_P B_W C_\sigma)^{L-\ell} C_\sigma L_{\text{loss}} E_H + U C_\sigma \sum_{i=0}^{L-\ell-1} (B_P B_W C_\sigma)^i + L_\sigma B_J E_Z \tag{7}$$

where $U = B_P(B_W B_J E_Z L_\sigma + B_{\Delta W} B_M + B_W B_{\Delta M})$. We prove them by induction as follows.

$$\|\widetilde{M}^{(t,\ell)} - M^{(\ell)}\|_F$$
$$= \|\widetilde{J}^{(t,\ell)} \circ \sigma'(\widetilde{Z}^{(t,\ell)}) - J^{(\ell)} \circ \sigma'(Z^{(\ell)})\|_F$$
$$\leq \|\widetilde{J}^{(t,\ell)} \circ \sigma'(\widetilde{Z}^{(t,\ell)}) - \widetilde{J}^{(t,\ell)} \circ \sigma'(Z^{(\ell)})\|_F + \|\widetilde{J}^{(t,\ell)} \circ \sigma'(Z^{(\ell)}) - J^{(\ell)} \circ \sigma'(Z^{(\ell)})\|_F$$
$$\leq B_J \|\sigma'(\widetilde{Z}^{(t,\ell)}) - \sigma'(Z^{(\ell)})\|_F + C_\sigma \|\widetilde{J}^{(t,\ell)} - J^{(\ell)}\|_F$$

Here $\|\sigma'(\widetilde{Z}^{(t,\ell)}) - \sigma'(Z^{(\ell)})\|_F \leq L_\sigma E_Z$. With Equation 6,

$$\|\widetilde{M}^{(t,\ell)} - M^{(\ell)}\|_F \leq (B_P B_W C_\sigma)^{L-\ell} C_\sigma L_{\text{loss}} E_H + U C_\sigma \sum_{i=0}^{L-\ell-1} (B_P B_W C_\sigma)^i + L_\sigma B_J E_Z$$

On the other hand,

$$\|\widetilde{J}^{(t,\ell-1)} - J^{(\ell-1)}\|_F$$
$$= \|P_{in}^\top \widetilde{M}^{(t,\ell)}[\widetilde{W}^{(t,\ell)}]^\top + P_{bd}^\top \widetilde{M}^{(t-1,\ell)}[\widetilde{W}^{(t-1,\ell)}]^\top - P^\top M^{(\ell)}[W^{(\ell)}]^\top\|_F$$
$$= \|P^\top(\widetilde{M}^{(t,\ell)} - M^{(\ell)})[W^{(\ell)}]^\top + P_{bd}^\top(\widetilde{M}^{(t-1,\ell)}[\widetilde{W}^{(t-1,\ell)}]^\top - \widetilde{M}^{(t,\ell)}[\widetilde{W}^{(t,\ell)}]^\top)\|_F$$
$$\leq \|P^\top(\widetilde{M}^{(t,\ell)} - M^{(\ell)})[W^{(\ell)}]^\top\|_F + \|P_{bd}^\top(\widetilde{M}^{(t-1,\ell)}[\widetilde{W}^{(t-1,\ell)}]^\top - \widetilde{M}^{(t,\ell)}[\widetilde{W}^{(t,\ell)}]^\top)\|_F$$
$$\leq B_P B_W \|\widetilde{M}^{(t,\ell)} - M^{(\ell)}\|_F + B_P \|\widetilde{M}^{(t-1,\ell)}[\widetilde{W}^{(t-1,\ell)}]^\top - \widetilde{M}^{(t,\ell)}[\widetilde{W}^{(t,\ell)}]^\top\|_F$$

The first part is bounded by Equation 7. For the second part,

$$\|\widetilde{M}^{(t-1,\ell)}[\widetilde{W}^{(t-1,\ell)}]^\top - \widetilde{M}^{(t,\ell)}[\widetilde{W}^{(t,\ell)}]^\top\|_F$$
$$\leq \|\widetilde{M}^{(t-1,\ell)}[\widetilde{W}^{(t-1,\ell)}]^\top - \widetilde{M}^{(t-1,\ell)}[\widetilde{W}^{(t,\ell)}]^\top\|_F + \|\widetilde{M}^{(t-1,\ell)}[\widetilde{W}^{(t,\ell)}]^\top - \widetilde{M}^{(t,\ell)}[\widetilde{W}^{(t,\ell)}]^\top\|_F$$
$$\leq B_{\Delta W} B_M + B_W B_{\Delta M}$$

Therefore,

$$\|\widetilde{J}^{(t,\ell-1)} - J^{(\ell-1)}\|_F$$
$$\leq B_P B_W \|\widetilde{M}^{(t,\ell)} - M^{(\ell)}\|_F + B_P \|\widetilde{M}^{(t-1,\ell)}[\widetilde{W}^{(t-1,\ell)}]^\top - \widetilde{M}^{(t,\ell)}[\widetilde{W}^{(t,\ell)}]^\top\|_F$$
$$\leq (B_P B_W C_\sigma)^{L-\ell+1} L_{\text{loss}} E_H + U \sum_{i=1}^{L-\ell} (B_P B_W C_\sigma)^i + U$$
$$= (B_P B_W C_\sigma)^{L-\ell+1} L_{\text{loss}} E_H + U \sum_{i=0}^{L-\ell} (B_P B_W C_\sigma)^i$$

$\square$

**Lemma A.8.** $\|\widetilde{G}^{(t,\ell)} - G^{(\ell)}\|_F \leq E_G$ *where* $E_G = B_P(B_H E_M + B_M E_H)$

*Proof.*

$$\|\widetilde{G}^{(t,\ell)} - G^{(\ell)}\|_F$$
$$= \left\|[P_{in}\widetilde{H}^{(t,\ell-1)} + P_{bd}\widetilde{H}^{(t-1,\ell-1)}]^\top \widetilde{M}^{(t,\ell)} - [PH^{(\ell)}]^\top M^{(\ell)}\right\|_F$$
$$\leq \left\|[P_{in}\widetilde{H}^{(t,\ell-1)} + P_{bd}\widetilde{H}^{(t-1,\ell-1)}]^\top \widetilde{M}^{(t,\ell)} - [PH^{(\ell-1)}]^\top \widetilde{M}^{(t,\ell)}\right\|_F$$
$$+ \left\|[PH^{(\ell-1)}]^\top \widetilde{M}^{(t,\ell)} - [PH^{(\ell-1)}]^\top M^{(\ell)}\right\|_F$$
$$\leq B_M(\|P(\widetilde{H}^{(t,\ell-1)} - H^{(\ell-1)}) + P_{bd}(\widetilde{H}^{(t-1,\ell-1)} - \widetilde{H}^{(t,\ell-1)})\|_F) + B_P B_H E_M$$
$$\leq B_M B_P(E_H + B_{\Delta H}) + B_P B_H E_M$$

$\square$

By summing up from $\ell = 1$ to $\ell = L$ to both sides, we have

**Corollary A.9.** $\|\nabla\widetilde{\mathcal{L}}(\theta) - \nabla\mathcal{L}(\theta)\|_2 \leq E_{loss}$ where $E_{loss} = LE_G$.

According to the derivation of $E_{\text{loss}}$, we observe that $E_{\text{loss}}$ contains a factor $\eta$. To simplify the expression of $E_{\text{loss}}$, we assume that $B_P B_W C_\sigma \leq \frac{1}{2}$ without loss of generality, and rewrite Corollary A.9 as the following.

**Corollary A.10.** $\|\nabla\widetilde{\mathcal{L}}(\theta) - \nabla\mathcal{L}(\theta)\|_2 \leq \eta E$ where

$$E = \frac{1}{8}LB_P^3 B_X^2 C_{loss} C_\sigma \left(3B_X C_\sigma^2 L_{loss} + 6B_X C_{loss} L_\sigma + 10C_{loss} C_\sigma^2\right)$$

### A.4 Proof of the Main Theorem

We first introduce a lemma before the proof of our main theorem.

**Lemma A.11** (Lemma 1 in (Cong et al., 2021)). *An $L$-layer GCN is $L_f$-Lipschitz smoothness, i.e.,* $\|\nabla\mathcal{L}(\theta_1) - \nabla\mathcal{L}(\theta_2)\|_2 \leq L_f\|\theta_1 - \theta_2\|_2$.

Now we prove the main theorem.

**Theorem A.12** (Convergence of PipeGCN, formal). *Under Assumptions A.1, A.2, and A.3, we can derive the following by choosing a learning rate $\eta = \frac{\sqrt{\varepsilon}}{E}$ and number of training iterations $T = (\mathcal{L}(\theta^{(1)}) - \mathcal{L}(\theta^*))E\varepsilon^{-\frac{3}{2}}$:*

$$\frac{1}{T}\sum_{t=1}^{T}\|\nabla\mathcal{L}(\theta^{(t)})\|_2 \leq 3\varepsilon$$

*where $E$ is defined in Corollary A.10, $\varepsilon > 0$ is an arbitrarily small constant, $\mathcal{L}(\cdot)$ is the loss function, $\theta^{(t)}$ and $\theta^*$ represent the parameter vector at iteration $t$ and the optimal parameter respectively.*

*Proof.* With the smoothness of the model,

$$\mathcal{L}(\theta^{(t+1)}) \leq \mathcal{L}(\theta^{(t)}) + \left\langle\nabla\mathcal{L}(\theta^{(t)}), \theta^{(t+1)} - \theta^{(t)}\right\rangle + \frac{L_f}{2}\|\theta^{(t+1)} - \theta^{(t)}\|_2^2$$

$$= \mathcal{L}(\theta^{(t)}) - \eta\left\langle\nabla\mathcal{L}(\theta^{(t)}), \nabla\widetilde{\mathcal{L}}(\theta^{(t)})\right\rangle + \frac{\eta^2 L_f}{2}\|\nabla\widetilde{\mathcal{L}}(\theta^{(t)})\|_2^2$$

Let $\delta^{(t)} = \nabla\widetilde{\mathcal{L}}(\theta^{(t)}) - \nabla\mathcal{L}(\theta^{(t)})$ and $\eta \leq 1/L_f$, we have

$$\mathcal{L}(\theta^{(t+1)}) \leq \mathcal{L}(\theta^{(t)}) - \eta\left\langle\nabla\mathcal{L}(\theta^{(t)}), \nabla\mathcal{L}(\theta^{(t)}) + \delta^{(t)}\right\rangle + \frac{\eta}{2}\|\nabla\mathcal{L}(\theta^{(t)}) + \delta^{(t)}\|_2^2$$

$$\leq \mathcal{L}(\theta^{(t)}) - \frac{\eta}{2}\|\nabla\mathcal{L}(\theta^{(t)})\|_2^2 + \frac{\eta}{2}\|\delta^{(t)}\|_2^2$$

From Corollary A.10 we know that $\|\delta^{(t)}\|_2 < \eta E$. After rearranging the terms,

$$\|\nabla\mathcal{L}(\theta^{(t)})\|_2^2 \leq \frac{2}{\eta}(\mathcal{L}(\theta^{(t)}) - \mathcal{L}(\theta^{(t+1)})) + \eta^2 E^2$$

Summing up from $t = 1$ to $T$ and taking the average,

$$\frac{1}{T}\sum_{t=1}^{T}\|\nabla\mathcal{L}(\theta^{(t)})\|_2^2 \leq \frac{2}{\eta T}(\mathcal{L}(\theta^{(1)}) - \mathcal{L}(\theta^{(T+1)})) + \eta^2 E^2$$

$$\leq \frac{2}{\eta T}(\mathcal{L}(\theta^{(1)}) - \mathcal{L}(\theta^*)) + \eta^2 E^2$$

where $\theta^*$ is the minimum point of $\mathcal{L}(\cdot)$. By taking $\eta = \frac{\sqrt{\varepsilon}}{E}$ and $T = (\mathcal{L}(\theta^{(1)}) - \mathcal{L}(\theta^*))E\varepsilon^{-\frac{3}{2}}$ with an arbitrarily small constant $\varepsilon > 0$, we have

$$\frac{1}{T}\sum_{t=1}^{T}\|\nabla\mathcal{L}(\theta^{(t)})\|_2 \leq 3\varepsilon$$

$\square$

# B  TRAINING TIME BREAKDOWN OF FULL-GRAPH TRAINING METHODS

To understand why PipeGCN significantly boosts the training throughput over full-graph training methods, we provide the detailed time breakdown in Tab. 6 using the same model as Tab. 3 (4-layer GraphSAGE, 256 hidden units), in which "GCN" denotes the vanilla partition-parallel training illustrated in Fig. 1(a). We observe that PipeGCN greatly saves communication time.

Table 6: Epoch time breakdown of full-graph training methods on the Reddit dataset.

| Method | Total time (s) | Compute (s) | Communication (s) | Reduce (s) |
|---|---|---|---|---|
| ROC (2 GPUs) | 3.63 | 0.5 | 3.13 | 0.00 |
| CAGNET (c=1, 2 GPUs) | 2.74 | 1.91 | 0.65 | 0.18 |
| CAGNET (c=2, 2 GPUs) | 5.41 | 4.36 | 0.09 | 0.96 |
| GCN (2 GPUs) | 0.52 | 0.17 | 0.34 | 0.01 |
| PipeGCN (2 GPUs) | 0.27 | 0.25 | 0.00 | 0.02 |
| ROC (4 GPUs) | 3.34 | 0.42 | 2.92 | 0.00 |
| CAGNET (c=1, 4 GPUs) | 2.31 | 0.97 | 1.23 | 0.11 |
| CAGNET (c=2, 4 GPUs) | 2.26 | 1.03 | 0.55 | 0.68 |
| GCN (4 GPUs) | 0.48 | 0.07 | 0.40 | 0.01 |
| PipeGCN (4 GPUs) | 0.23 | 0.10 | 0.10 | 0.03 |

## C  TRAINING TIME IMPROVEMENT BREAKDOWN OF PIPEGCN

To understand the training time improvement offered by PipeGCN, we further breakdown the epoch time into three parts (intra-partition computation, inter-partition communication, and reduce for aggregating model gradient) and provide the result in Fig. 8. We can observe that: 1) inter-partition **communication dominates the training time** in vanilla partition-parallel training (GCN); 2) **PipeGCN (with or without smoothing) greatly hides the communication overhead** across different number of partitions and all datasets, e.g., the communication time is hidden completely in 2-partition Reddit and almost completely in 3-partition Yelp, thus the substantial reduction in training time; and 3) the proposed **smoothing incurs only minimal overhead** (i.e., minor difference between PipeGCN and PipeGCN-GF). Lastly, we also notice that *when communication ratio is extremely large (85%+), PipeGCN hides communication significantly but not completely* (e.g., 10-partition ogbn-products), in which case we can employ those compression and quantization techniques (Alistarh et al. (2017); Seide et al. (2014); Wen et al. (2017); Li et al. (2018a); Yu et al. (2018)) from the area of general distributed SGD for further reducing the communication, as the compression is orthogonal to the pipeline method. Besides compression, we can also increase the pipeline depth of PipeGCN, e.g., using two iterations of compute to hide one iteration of communication, which is left to our future work.

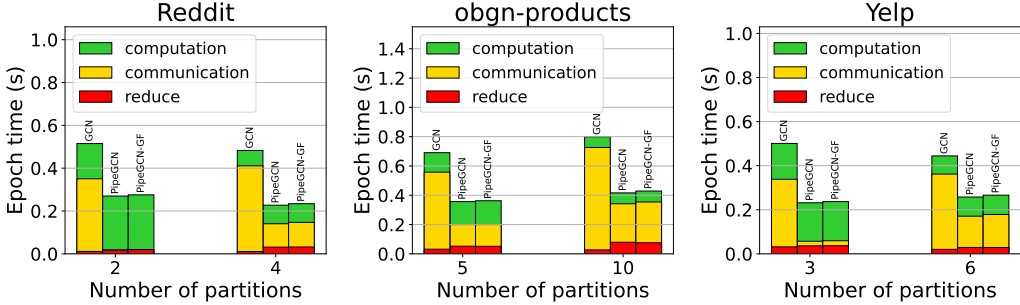

Figure 8: Training time breakdown of vanilla partition-parallel training (GCN), PipeGCN, and PipeGCN with smoothing (PipeGCN-GF).

## D MAINTAINING CONVERGENCE SPEED (ADDITIONAL EXPERIMENTS)

We provide the additional convergence curves on Yelp in Fig. 9. We can see that ***PipeGCN and its variants maintain the convergence speed w.r.t the number of epochs while substantially reducing the end-to-end training time.***

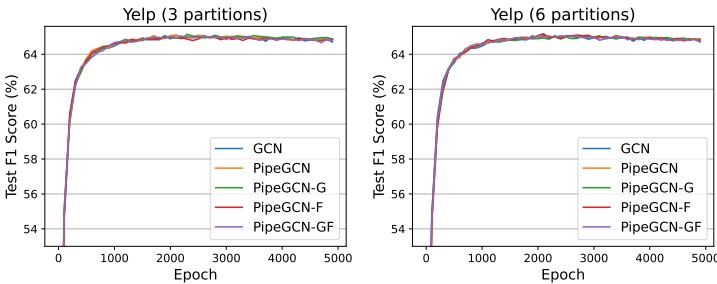

Figure 9: The epoch-to-accuracy comparison on "Yelp" among the vanilla partition-parallel training (GCN) and PipeGCN variants (PipeGCN*), where *PipeGCN and its variants achieve a similar convergence as the vanilla training (without staleness) but are twice as fast in terms of wall-clock time* (see the Throughput improvement in Tab. 4 of the main content).

# E   SCALING GCN TRAINING OVER MULTIPLE GPU SERVERS

We also scale up PipeGCN training over multiple GPU servers (each contains AMD Radeon Instinct MI60 GPUs, an AMD EPYC 7642 CPU, and 48 lane PCI 3.0 connecting CPU-GPU and GPU-GPU) networked with 10Gbps Ethernet.

The accuracy results of PipeGCN and its variants are summarized in Tab. 7:

Table 7: The accuracy of PipeGCN and its variants on Reddit.

| #partitions (#node×#gpus) | PipeGCN | PipeGCN-F | PipeGCN-G | PipeGCN-GF |
|---|---|---|---|---|
| 2 (1×2) | 97.12% | 97.09% | 97.14% | 97.12% |
| 3 (1×3) | 97.01% | 97.15% | 97.17% | 97.14% |
| 4 (1×4) | 97.04% | 97.10% | 97.09% | 97.10% |
| 6 (2×3) | 97.09% | 97.12% | 97.08% | 97.10% |
| 8 (2×4) | 97.02% | 97.06% | 97.15% | 97.03% |
| 9 (3×3) | 97.03% | 97.08% | 97.11% | 97.08% |
| 12 (3×4) | 97.05% | 97.05% | 97.12% | 97.10% |
| 16 (4×4) | 96.99% | 97.02% | 97.14% | 97.12% |

Furthermore, we provide PipeGCN's speedup against vanilla partition-parallel training in Tab. 8:

Table 8: The speedup of PipeGCN and its vatiants against vanilla partition-parallel training on Reddit.

| #nodes×#gpus | GCN | PipeGCN | PipeGCN-G | PipeGCN-F | PipeGCN-GF |
|---|---|---|---|---|---|
| 1×2 | 1.00× | 1.16× | 1.16× | 1.16× | 1.16× |
| 1×3 | 1.00× | 1.22× | 1.22× | 1.22× | 1.22× |
| 1×4 | 1.00× | 1.29× | 1.28× | 1.29× | 1.28× |
| 2×2 | 1.00× | 1.61× | 1.60× | 1.61× | 1.60× |
| 2×3 | 1.00× | 1.64× | 1.64× | 1.64× | 1.64× |
| 2×4 | 1.00× | 1.41× | 1.42× | 1.41× | 1.37× |
| 3×2 | 1.00× | 1.65× | 1.65× | 1.65× | 1.65× |
| 3×3 | 1.00× | 1.48× | 1.49× | 1.50× | 1.48× |
| 3×4 | 1.00× | 1.35× | 1.36× | 1.35× | 1.34× |
| 4×2 | 1.00× | 1.64× | 1.63× | 1.63× | 1.62× |
| 4×3 | 1.00× | 1.38× | 1.38× | 1.38× | 1.38× |
| 4×4 | 1.00× | 1.30× | 1.29× | 1.29× | 1.29× |

From the two tables above, we can observe that our PipeGCN family consistently **maintains the accuracy** of the full-graph training, while **improving the throughput by 15%∼66%** regardless of the machine settings and number of partitions.

## F  IMPLEMENTATION DETAILS

We discuss the details of the effective and efficient implementation of PipeGCN in this section.

First, for parallel communication and computation, a second cudaStream is required for communication besides the default cudaStream for computation. To also save memory buffers for communication, we batch all communication (e.g., from different layers) into this second cudaStream. When the popular communication backend, Gloo, is used, we parallelize the CPU-GPU transfer with CPU-CPU transfer.

Second, when Dropout layer is used in GCN model, it should be applied after communication. The implementation of the dropout layer for PipeGCN should be considered carefully so that the dropout mask remains consistent for the input tensor and corresponding gradient. If the input feature passes through the dropout layer before being communicated, during the backward phase, the dropout mask is changed and the gradient of masked values is involved in the computation, which introduces noise to the calculation of followup gradients. As a result, the dropout layer can only be applied after receiving boundary features.

