# OpenReview forum: "PipeGCN: Efficient Full-Graph Training of Graph Convolutional Networks with Pipelined Feature Communication"
_ICLR.cc/2022/Conference — ICLR 2022 Poster_

### Official Review · Reviewer_L5WF · 2021-10-20

**Correctness:** 3
**Technical Novelty And Significance:** 3
**Empirical Novelty And Significance:** 3
**Recommendation:** 6
**Confidence:** 4

**Main Review:**

Strengths
- Simple method that is relatively easy to implement compared to other parallel GNN systems. Does not require complex memory management (e.g. involving CPU memory). That makes it potentially easy to extend to a distributed network setting (multiple machines).
- The staleness is in activations/gradients, not weights. Intuitively, this leads to more stable convergence (vs stale weights) because (1) the stale activations/gradients only occur on the partition boundaries; (2) the activation/gradient for a neuron on the boundary is a combination of stale (inter-partition) and non-stale (intra-partition) activations/gradients
- I did not evaluate the theoretical proofs or convergence rate for mathematical correctness, but the fact that the method converges at all is intuitively reasonable, given that stale gradient systems are already known to converge.

Weaknesses
- It is rather surprising that vanilla partition parallel, a simple baseline that many other GNN systems are adapted from, beats recently published (2020) methods by a very wide margin, such as ROC and CAGNET. Although the authors did give some analysis as to why this is the case, I am not convinced those could be the major factors. It is more likely that the baselines have been misconfigured. For the avoidance of doubt, the authors should provide numerical evidence that their obtained results are consistent with the results reported in the ROC and CAGNET papers.
- The method relies on a somewhat brittle assumption that inter-partition communication time is roughly equal to activation/gradient computation time. If the communication time for a given model and partitioning were significantly smaller or larger than the computation time, the throughput gain from parallelizing computation with communication would be much less impressive. Ultimately, the method does not truly solve the issue of overwhelming communication volume - which sampling-based methods do address. The paper should acknowledge this limitation, rather than claiming superiority in all scenarios to sampling based methods.

Minor issues and suggestions
- The authors should not use the word "distributed" unless the implementation and experiments support networked machines. The hardware configuration in the experiments is a single machine with multiple GPUs.
- It is interesting that the experiments were performed on commodity hardware with what I understand to be no NVLink, and only PCIe v3 x16 connections. The bandwidth of PCIe v3 connections is roughly comparable to 100-200Gbps network connections. I take that as a promising sign that the method will continue to perform well with distributed network machines.


**Summary Of The Paper:**

A method of partitioning GNNs, and using stale versions of activations across partitions. This allows activations to be communicated in parallel (rather than sequentially) across GPUs, which improves throughput. Theoretical results show that the stale activations still result in convergence, and empirically the method appears to perform better than the rate given by the theory.

**Summary Of The Review:**

While the idea of stale activations/gradients is not new (as the authors have acknowledged in the related work), the convergence result seems new to the best of my limited knowledge. I am positive that (1) validation accuracy is essentially equal to the non-stale version; (2) based on the PCIe bandwidth in the experiments, the method should also work in distributed inifiniband or 100+Gbps ethernet settings. However, I do have concerns about (A) how the ROC and CAGNET results were obtained, and (B) the method only performs well in a narrow "Goldilocks zone" where computation time is roughly equal to communication time.

On the last point - the authors might want to consider what happens at the billion+ node scale. In my experience, the distribution of very high degree nodes increases, and it is likely that nodes with an extreme number of neighbors will end up on the boundary. This could greatly increase communication time relative to computation, and the method would no longer perform well.

Overall, it is an interesting if slightly flawed paper, and I lean towards acceptance.

---

> ### Author Response · Authors · 2021-11-21
> **Response to Reviewer L5WF: Part 1**
>
> > Overall, it is an interesting if slightly flawed paper, and I lean towards acceptance.
>
> > **Simple** method that is relatively easy to implement compared to other parallel GNN systems. Does not require complex memory management (e.g. involving CPU memory). That makes it potentially easy to extend to a distributed network setting (multiple machines).
>
> > The **staleness is in activations/gradients, not weights**. Intuitively, this leads to more stable convergence (vs stale weights) because (1) the stale activations/gradients only occur on the partition boundaries; (2) the activation/gradient for a neuron on the boundary is a combination of stale (inter-partition) and non-stale (intra-partition) activations/gradients.
>
> > I did not evaluate the theoretical proofs or convergence rate for mathematical correctness, but the fact that the method **converges** at all is **intuitively reasonable**, given that stale gradient systems are already known to converge.
>
> > It is interesting that the experiments were performed on commodity hardware[ with what I understand to be no NVLink, and only PCIe v3 x16 connections. The bandwidth of PCIe v3 connections is roughly comparable to 100-200Gbps network connections. **I take that as a promising sign that the method will continue to perform well with distributed network machines**.
>
> Thank you for your deep understanding and constructive review! Especially thanks for the appreciation of our PipeGCN’s simplicity, extensibility, staleness in features (not weights), theoretical convergence proof, as well as necessitating only commodity hardware.
>
> > It is rather surprising that vanilla partition parallel, a simple baseline that many other GNN systems are adapted from, beats recently published (2020) methods by a very wide margin, such as ROC and CAGNET. Although the authors did give some analysis as to why this is the case, I am not convinced those could be the major factors. It is more likely that the **baselines have been misconfigured**. For the avoidance of doubt, **the authors should provide numerical evidence that their obtained results are consistent with the results reported in the ROC and CAGNET papers**.
>
> Great question. We have carefully checked the configurations of our baselines (such as ROC [1] and CAGNET [2]), followed the setting in their papers, and even contacted some of their authors for correctness confirmation. Furthermore, we have trained the model of ROC (2-layer GCN, 256 hidden units) and can reproduce their reported accuracy of 95.4% on Reddit, which is consistent with Figure 7 of ROC [1].
>
> To further address your concern, we profile the total communication volume (i.e., the major bottleneck in distributed GCN training) by using the standard tool `nvprof` [3], and list the results for each epoch on Reddit below:
>
> |                                 | 2 GPUs  | 4 GPUs  |
> | ------------------------------- | ------- | ------- |
> | ROC                             | 35.5 GB | 38.3 GB |
> | CAGNET (c=1)                    | 13.8 GB | 16.3 GB |
> | CAGNET (c=2)                    | 15.1 GB | 16.4 GB |
> | Vanilla  Partition-Parallel GCN | 1.7 GB  | 3.5 GB  |
>
>
>
> As can be observed from the above table, the wide winning margin of vanilla partition-parallel GCN over ROC and CAGNET can be attributed to the significant gap between their **communication volumes** (i.e., an order-of-magnitude gap). This validates our analysis in the submitted manuscript -- ROC requires repeated CPU-GPU swapping for each sub-partition to compute on GPUs, while CAGNET requires redundant communication by broadcasting all features.
>
> > The method relies on a somewhat **brittle assumption** that inter-partition **communication time is roughly equal to** activation/gradient **computation time**. If the communication time for a given model and partitioning were significantly smaller or larger than the computation time, the throughput gain from parallelizing computation with communication would be much less impressive.
>
> > the method only performs well in a narrow **"Goldilocks zone"** where **computation time is roughly equal to communication time**.
>
> Great perspective. We humbly clarify that PipeGCN does NOT rely on the assumption of equal communication-computation time. This is because our communication of each layer is **NOT** serialized and then hidden by computation, but instead we **communicate all layers in parallel** and let them be hidden by computation (see Figure 2(b)), such that even small compute time can hide large communication time in total.
>
> As validated in our experiments, the communication can take up to 90% total training time. For instance, in Figure 10 (Appendix D), the vanilla 4-layer GCN takes 83% time for communication, but PipeGCN still improves the efficiency by more than 100%.

---

> > ### Author Response · Authors · 2021-11-21
> > **Response to Reviewer L5WF: Part 2**
> >
> > > Ultimately, the method **does not truly solve the issue of overwhelming communication volume** - which sampling-based methods do address. The paper should acknowledge this limitation, rather than claiming superiority in all scenarios to sampling based methods.
> >
> > Great suggestion. Indeed, PipeGCN does not reduce communication volume but instead hides it, which is to address the overwhelming communication overhead from a different angle. As our approach is orthogonal to sampling-based methods, they can be combined together for further improvement.
> >
> > > **The authors should not use the word "distributed" unless the implementation and experiments support networked machines.** The hardware configuration in the experiments is a single machine with multiple GPUs.
> >
> > Great suggestion. Our implementation does support networked machines. Due to our limited resources as an academic group, we only evaluated a single machine setup during submission.
> >
> > As requested, we rent a large GPU cluster for further evaluations of PipeGCN on Reddit over multiple machine nodes (each contains AMD Radeon Instinct MI60 GPUs, an AMD EPYC 7642 CPU, 48 lane PCI 3.0 connecting CPU-GPU and GPU-GPU) with 10 Gbps Ethernet connecting all nodes.
> >
> > We are aware that this network bandwidth is limited (due to our limited budget) and decreases the performance of distributed GCN training and PipeGCN, but we still provide the results below:
> >
> > 1) the accuracy of PipeGCN and its variants:
> >
> > | #partitions (#node*#gpus) | PipeGCN | PipeGCN-F | PipeGCN-G | PipeGCN-GF |
> > | ------------------------- | ------- | --------- | --------- | ---------- |
> > | 2 (1*2)                   | 97.12%  | 97.09%    | 97.14%    | 97.12%     |
> > | 3 (1*3)                   | 97.01%  | 97.15%    | 97.17%    | 97.14%     |
> > | 4 (1*4)                   | 97.04%  | 97.10%    | 97.09%    | 97.10%     |
> > | 6 (2*3)                   | 97.09%  | 97.12%    | 97.08%    | 97.10%     |
> > | 8 (2*4)                   | 97.02%  | 97.06%    | 97.15%    | 97.03%     |
> > | 9 (3*3)                   | 97.03%  | 97.08%    | 97.11%    | 97.08%     |
> > | 12 (3*4)                  | 97.05%  | 97.05%    | 97.12%    | 97.10%     |
> > | 16 (4*4)                  | 96.99%  | 97.02%    | 97.14%    | 97.12%     |
> >
> >
> > 2) the speedup of PipeGCN against vanilla distributed GCN training:
> >
> > | #nodes*#gpus | GCN   | PipeGCN | PipeGCN-G | PipeGCN-F | PipeGCN-GF |
> > | ------------ | ----- | ------- | --------- | --------- | ---------- |
> > | 1*2          | 1.00x | 1.16x   | 1.16x     | 1.16x     | 1.16x      |
> > | 1*3          | 1.00x | 1.22x   | 1.22x     | 1.22x     | 1.22x      |
> > | 1*4          | 1.00x | 1.29x   | 1.28x     | 1.29x     | 1.28x      |
> > | 2*2          | 1.00x | 1.61x   | 1.60x     | 1.61x     | 1.60x      |
> > | 2*3          | 1.00x | 1.64x   | 1.64x     | 1.64x     | 1.64x      |
> > | 2*4          | 1.00x | 1.41x   | 1.42x     | 1.41x     | 1.37x      |
> > | 3*2          | 1.00x | 1.65x   | 1.65x     | 1.65x     | 1.65x      |
> > | 3*3          | 1.00x | 1.48x   | 1.49x     | 1.50x     | 1.48x      |
> > | 3*4          | 1.00x | 1.35x   | 1.36x     | 1.35x     | 1.34x      |
> > | 4*2          | 1.00x | 1.64x   | 1.63x     | 1.63x     | 1.62x      |
> > | 4*3          | 1.00x | 1.38x   | 1.38x     | 1.38x     | 1.38x      |
> > | 4*4          | 1.00x | 1.30x   | 1.29x     | 1.29x     | 1.29x      |
> >
> >
> >
> > From the above tables, we can observe that our PipeGCN family still **maintains the accuracy** of the full-graph training in distributed settings, while **improving the throughput by 15%~66%**, i.e., non-trivial considering the limited network bandwidth of only 10Gbps. We will add these results to the revision.
> >
> > > **The authors might want to consider what happens at the billion+ node scale**. In my experience, the distribution of very high degree nodes increases, and it is likely that nodes with an extreme number of neighbors will end up on the boundary. This could greatly increase communication time relative to computation, and the method would no longer perform well.
> >
> > Good perspective. As explained above, PipeGCN is able to reduce large communication overhead by hiding communication in parallel with only a small computation time, thus it may still work well at an extreme scale. In future, we plan to employ orthogonal techniques like sampling/quantization to further reduce the communication volume.
> >
> > To further address your concern, we increase the partition/GPU number of ogbn-papers100M from 4 to 32, and observe that the largest partition contains **2x** boundary nodes than the average, which satisfies your described condition. Even under this extreme case, PipeGCN still **improves the throughput by 60%** (0.096 epochs/s -> 0.154 epochs/s) on the rented AMD cluster.

---

> > > ### Author Response · Authors · 2021-11-21
> > > **Response to Reviewer L5WF: Reference**
> > >
> > > [1] Jia, Zhihao, et al. "Improving the accuracy, scalability, and performance of graph neural networks with roc." Proceedings of Machine Learning and Systems 2 (2020): 187-198.
> > >
> > > [2] Tripathy, Alok, Katherine Yelick, and Aydın Buluç. "Reducing communication in graph neural network training." SC20: International Conference for High Performance Computing, Networking, Storage and Analysis. IEEE, 2020.
> > >
> > > [3] Bradley, Thomas. "GPU performance analysis and optimisation." NVIDIA Corporation (2012).

---

> > > > ### Comment · Reviewer_L5WF · 2021-11-24
> > > > **Thank you for the responses**
> > > >
> > > > The authors have thoroughly addressed my questions. I accept their clarification on the computation vs communication time issue. The addition of distributed training results will definitely strengthen the paper, and I am also encouraged to see the authors' careful responses to the other reviewers' questions.
> > > >
> > > > I have changed my personal rating of the paper to 7 (up from my original score of 6) as I lean more towards acceptance. Unfortunately the ICLR system only allows me to enter 6 or 8 as scores, hence this number would not be reflected in the system.

---

### Official Review · Reviewer_59N6 · 2021-11-01

**Correctness:** 3
**Technical Novelty And Significance:** 2
**Empirical Novelty And Significance:** 3
**Recommendation:** 6
**Confidence:** 4

**Main Review:**

**Strengths**
1. It analyzes two efficiency bottlenecks in distributed training, including communication overhead and synchronization, and proposes PipeGCN to pipelining the inter-partition communication with intra-partition computation.
2. The paper provides novel theoretical proof of the convergence of GCN training with stale feature and feature gradients, which is useful for future work.
3. The experiment results show a significant speedup of the training for the GCN model on multiple reported datasets and compares it with the other latest methods.

**Weakness**
1. The PipeGCN aims at scaling graph neural network training, but the setup of the largest dataset, ogbn-papers100M, is not practical. With only 2 layers and 8 hidden units, the GNN may not learn anything from the graph. Also, the accuracy on ogbn-papers100M is missed in the result table. With that in mind, I am concerned about the scalability of the system.
2. For each dataset, the paper only shows the results for a fixed number of partitions. It would be great to see how the speed and convergence differ when the number of partitions and computation nodes increases.
3. The results in Table 6 are vague, without showing the dataset used and what is the dist GCN method.

**Summary Of The Paper:**

The authors propose PipeGCN - a method for efficiently distributed full-graph GCN training. The method pipelines communication with computation in distributed GCN training to hide substantial communication overhead. The paper leads an effort to study pipelined and asynchronous distributed training of GCNs with a new smoothing method aimed at reducing the error incurred by stale features/feature gradients at minimal overhead.

**Summary Of The Review:**

The paper proposes an interesting framework to speed up the GCN training with the theoretical proof for the complexity. However, as mentioned in the weaknesses above, the weird setup on the largest dataset, ogbn-papers100M, and lacking the evaluation on a different number of computation nodes, it is hard to understand the scalability of the system.

---

> ### Author Response · Authors · 2021-11-21
> **Response to Reviewer 59N6: Part 1**
>
> > **Strengths**
> > 1. It **analyzes two efficiency bottlenecks** in distributed training, including communication overhead and synchronization, and proposes PipeGCN to pipelining the inter-partition communication with intra-partition computation.
> > 2. The paper provides **novel theoretical proof of the convergence** of GCN training with stale feature and feature gradients, which is **useful for future work**.
> > 3. The experiment results show a **significant speedup** of the training for the GCN model on **multiple reported datasets** and compares it with the other **latest methods**.
>
> Thank you for your informative review! Especially thanks for the appreciation of our contributions on efficiency analysis, novel convergence proof, as well as extensive evaluations.
>
> > the weird setup on the largest dataset, ogbn-papers100M, and lacking the evaluation on a different number of computation nodes, it is **hard to understand the scalability of the system**.
>
> > The PipeGCN aims at scaling graph neural network training, but **the setup of the largest dataset, ogbn-papers100M, is not practical**. With only **2 layers** and **8 hidden units**, the GNN may not learn anything from the graph. Also, **the accuracy on ogbn-papers100M is missed** in the result table. With that in mind, I am concerned about the scalability of the system.
>
> As an academic research group, the 2 layer and 8 hidden units are set to accommodate our maximum hardware resources. Following your request, we rent a larger GPU cluster to scale the model to **3-layer** GraphSAGE with **48 hidden units**, and measure its training efficiency on this cluster with **32 GPUs** (AMD Radeon Instinct MI60) as show below:
>
>
> |            | Total Time | Comm Time | Reduce Time | Speedup |
> | ---------- | ---------- | --------- | ----------- | ------- |
> | GCN        | 10.5s      | 6.6s      | 1.2s        | 1.00x   |
> | PipeGCN    | 6.5s       | 2.6s      | 1.2s        | 1.60x   |
> | PipeGCN-GF | 6.7s       | 2.8s      | 1.1s        | 1.57x   |
>
>
> From the above table, we can see that even at such a large scale cluster where communication overhead dominates, PipeGCN and PipeGCN-GF still improve the training efficiency by 60% and 57% respectively compared to the GCN baseline. Please note that our GCN baseline is highly optimized and has already outperformed any state-of-the-art approach like ROC [1], CAGNET [2] (see Figure 3 in our paper). We will update our manuscript to include this set of experiments. Due to the resource limitation, we are still running the experiment to evaluate the accuracy of the vanilla model and PipeGCN, and will update you once the experiment is completed.
>
> Finally, we would like to humbly point out  that few related works have ever trained such a large GCN at this cluster scale. In particular, most related works (e.g., NeuGraph [3], ROC [1], Dorylus [4], $P^3$ [5], and GNNAdvisor [6]) only evaluated the throughput of a **2-layer GCN** while **ignoring its accuracy** for giant graphs. Specifically, $P^3$ [5] uses a 2-layer GCN with 32 hidden dimensions for this ogbn-papers100M dataset, which is less practical than our experiment setting above.

---

> > ### Author Response · Authors · 2021-11-21
> > **Response to Reviewer 59N6: Part 2**
> >
> >
> > > For each dataset, the paper only shows the results for a fixed number of partitions. It would be great to see **how the speed and convergence differ when the number of partitions and computation nodes increases**.
> >
> > Great suggestion! Our implementation does support multiple machines. Following your suggestion, we evaluate PipeGCN on Reddit over different number of partitions and multiple nodes (each contains AMD Radeon Instinct MI60 GPUs, an AMD EPYC 7642 CPU, and 48 lane PCI 3.0 connecting CPU-GPU and GPU-GPU) networked with 10Gbps Ethernet.
> >
> > The corresponding accuracy results of PipeGCN and its variants are summarized below:
> >
> > | #partitions (#node*#gpus) | PipeGCN | PipeGCN-F | PipeGCN-G | PipeGCN-GF |
> > | ------------------------- | ------- | --------- | --------- | ---------- |
> > | 2 (1*2)                   | 97.12%  | 97.09%    | 97.14%    | 97.12%     |
> > | 3 (1*3)                   | 97.01%  | 97.15%    | 97.17%    | 97.14%     |
> > | 4 (1*4)                   | 97.04%  | 97.10%    | 97.09%    | 97.10%     |
> > | 6 (2*3)                   | 97.09%  | 97.12%    | 97.08%    | 97.10%     |
> > | 8 (2*4)                   | 97.02%  | 97.06%    | 97.15%    | 97.03%     |
> > | 9 (3*3)                   | 97.03%  | 97.08%    | 97.11%    | 97.08%     |
> > | 12 (3*4)                  | 97.05%  | 97.05%    | 97.12%    | 97.10%     |
> > | 16 (4*4)                  | 96.99%  | 97.02%    | 97.14%    | 97.12%     |
> >
> > Furthermore, we provide the speedup against vanilla distributed GCN training below:
> >
> > | #nodes*#gpus | GCN   | PipeGCN | PipeGCN-G | PipeGCN-F | PipeGCN-GF |
> > | ------------ | ----- | ------- | --------- | --------- | ---------- |
> > | 1*2          | 1.00x | 1.16x   | 1.16x     | 1.16x     | 1.16x      |
> > | 1*3          | 1.00x | 1.22x   | 1.22x     | 1.22x     | 1.22x      |
> > | 1*4          | 1.00x | 1.29x   | 1.28x     | 1.29x     | 1.28x      |
> > | 2*2          | 1.00x | 1.61x   | 1.60x     | 1.61x     | 1.60x      |
> > | 2*3          | 1.00x | 1.64x   | 1.64x     | 1.64x     | 1.64x      |
> > | 2*4          | 1.00x | 1.41x   | 1.42x     | 1.41x     | 1.37x      |
> > | 3*2          | 1.00x | 1.65x   | 1.65x     | 1.65x     | 1.65x      |
> > | 3*3          | 1.00x | 1.48x   | 1.49x     | 1.50x     | 1.48x      |
> > | 3*4          | 1.00x | 1.35x   | 1.36x     | 1.35x     | 1.34x      |
> > | 4*2          | 1.00x | 1.64x   | 1.63x     | 1.63x     | 1.62x      |
> > | 4*3          | 1.00x | 1.38x   | 1.38x     | 1.38x     | 1.38x      |
> > | 4*4          | 1.00x | 1.30x   | 1.29x     | 1.29x     | 1.29x      |
> >
> > From the two tables above, we can observe that our PipeGCN family consistently **maintains the accuracy** of the full-graph training, while **improving the throughput by 15%~66%** regardless of the machine settings and number of partitions, which is consistent with our paper and will be added to the revision.
> >
> > > **The results in Table 6 are vague**, without showing the dataset used and what is the dist GCN method.
> >
> > Thanks for this valuable feedback. For Table 6, we study the training time breakdown on Reddit with different approaches training the same model in Table 3 (4-layer GraphSAGE, 256 hidden units). The ‘Dist GCN’ is the vanilla distributed GCN training illustrated in Figure 1 (a). We will provide this information in our revision.

---

> > > ### Author Response · Authors · 2021-11-21
> > > **Response to Reviewer 59N6: Reference**
> > >
> > > [1] Jia, Zhihao, et al. "Improving the accuracy, scalability, and performance of graph neural networks with roc." Proceedings of Machine Learning and Systems 2 (2020): 187-198.
> > >
> > > [2] Tripathy, Alok, Katherine Yelick, and Aydın Buluç. "Reducing communication in graph neural network training." SC20: International Conference for High Performance Computing, Networking, Storage and Analysis. IEEE, 2020.
> > >
> > > [3] Ma, Lingxiao, et al. "Neugraph: parallel deep neural network computation on large graphs." 2019 {USENIX} Annual Technical Conference ({USENIX}{ATC} 19). 2019. APA
> > >
> > > [4] Thorpe, John, et al. "Dorylus: Affordable, Scalable, and Accurate GNN Training over Billion-Edge Graphs." arXiv preprint arXiv:2105.11118 (2021).
> > >
> > > [5] Gandhi, Swapnil, and Anand Padmanabha Iyer. "P3: Distributed Deep Graph Learning at Scale." 15th {USENIX} Symposium on Operating Systems Design and Implementation ({OSDI} 21). 2021.
> > >
> > > [6] Wang, Yuke, et al. "GNNAdvisor: An Adaptive and Efficient Runtime System for {GNN} Acceleration on GPUs." 15th {USENIX} Symposium on Operating Systems Design and Implementation ({OSDI} 21). 2021.

---

> > > > ### Author Response · Authors · 2021-11-23
> > > > **We’d like to update the promised experiments**
> > > >
> > > > We’d like to update the ongoing experiments on the ogbn-papers100M dataset. Based on the best resource (32 MI60 GPUs) we have, we trained a 3-layer GraphSAGE with 48 hidden units. The accuracies of the vanilla model and PipeGCN are listed below:
> > > >
> > > > | Model      | Accuracy |
> > > > | ---------- | -------- |
> > > > | GCN        | 60.4%    |
> > > > | PipeGCN    | 60.2%    |
> > > > | PipeGCN-GF | 60.7%    |
> > > >
> > > > Due to the time limitation of the rebuttal period, we did not carefully tune the best hyper-parameters. We believe that the accuracies can be further improved, and will keep on improving this experiment.

---

> > > > > ### Comment · Reviewer_59N6 · 2021-11-30
> > > > > **Thank you for the detailed responses**
> > > > >
> > > > > I really appreciate the authors' detailed reply. It is great to see the #partition vs convergence/speed and the real speedup on the ogbn-papers dataset with the larger model. The reply solves most of my concerns and I would like to increase my score to 6, though I am still concerned about the experiments on ogbn-papers dataset as the network used is far from the SOTA. I really would like to see the performance of the proposed method on a really large and computation-intensive GNN (1024-dim hidden layer, as in the leaderboard) on powerful Nvidia GPUs, such as V100 in the final version.

---

> > > > > > ### Author Response · Authors · 2021-11-30
> > > > > > **Really appreciated for your reply.**
> > > > > >
> > > > > > Really appreciated for your reply. It is indeed of great value to further scale the model to 1024-dim with our proposed methods. As your request, we are actively conducting this experiment now. However, due to the closing window of rebuttal (in 6 hours), it is likely that our experiment cannot be finished on time, especially considering the massive resources required. According to our calculation, training such a large GCN requires 384/512 V100(32GB) and with a decent networking, for which we are still searching. Nonetheless, we will keep pushing this experiment and put it in the final versions as possible. Thanks again.

---

### Official Review · Reviewer_cPnX · 2021-11-02

**Correctness:** 4
**Technical Novelty And Significance:** 3
**Empirical Novelty And Significance:** 3
**Recommendation:** 6
**Confidence:** 3

**Main Review:**

I think the paper is a well-executed work: the discussions about related works are extensive, the idea behind PipeGCN is clearly explained, the convergence speed of PipeGCN is analyzed theoretically and the experiments are comprehensive.
I have some concerns on the novelty of the paper. As mentioned by the authors, Dorylus considered using stale features. To my knowledge, using stale gradients is well studied in Pipe-SGD related works. PipeGCN seems to combine the two ideas by using both stale features and stale gradients. Thus, the core problem is the technical contribution of the analysis of PipeGCN, which I do not have the technical expertise to evaluate.
I think the paper can be improved by fixing the following problems.
1. Experimental comparisons with Dorylus may be included to show the benefits of using stale gradients.
2. Figure 1 and Figure 2 can be improved. In Figure 1(b), the “…” for Part 2 should be removed and all communication tasks should be explicitly listed (e.g., communicate features, communicate gradients, and note that communicate gradients happen after some computation). In Figure 1(b) and (c), the computation and communication tasks should have iteration counts such that the pipelining idea can be understood. In Figure 2, “communicate for Next L1 backward” happens before the L1 backward of the current iteration, how is that possible? Is it a typo?
3. The influence of the smoothing technique can be better analyzed. In Table 4, there is not a consistent winner among PipeGCN, PipeGCN-G and PipeGCG-F in model accuracy.


**Summary Of The Paper:**

This paper proposes PipeGCN, which pipelines communication and computation in distributed GCN training to improve training throughput. Analysis is conducted to show the convergence speed when using both stale gradient and feature vectors. Extensive experiments are conducted to show that PipeGCN significantly improves the efficiency of vanilla distributed GCN training without hurting model accuracy.

**Summary Of The Review:**

The paper is complete but lacks comparison with an important related work and a justification of the technical challenges.

---

> ### Author Response · Authors · 2021-11-21
> **Response to Reviewer cPnX: Part 1**
>
> > The discussions about related works are extensive, the idea behind PipeGCN is clearly explained, the convergence speed of PipeGCN is analyzed theoretically and the experiments are comprehensive.
>
> Thank you for your careful and informative review! Especially thanks for the appreciation of our efforts on related works studies, idea explanation, convergence proof, as well as extensive evaluations.
>
> Before answering your questions, we humbly clarify one potential misunderstanding. This work surrounds stale feature and **stale feature gradient** but does NOT consider **stale weight gradient** as both Pipe-SGD [1] and its related works did. **Feature gradient** and **weight gradient** are fundamentally different in terms of both convergence and efficiency, and the former can lead to more stable convergence as pointed out by *Reviewer L5WF*.
>
>
> > I have some **concerns on the novelty of the paper**. As mentioned by the authors, Dorylus considered using **stale features**. To my knowledge, using **stale gradients** is well studied in Pipe-SGD related works. **PipeGCN seems to combine the two ideas** by using both stale features and stale gradients. Thus, the core problem is the technical contribution of the analysis of PipeGCN, which I do not have the technical expertise to evaluate.
>
> As humbly clarified above, this work is not about stale **weight** gradient as Pipe-SGD [1] related works did, instead it adopts both stale **feature** and **feature gradient** which differ distributed GCN training from classic distributed DNN training. Thus, in terms of novelty, our work is the first to provide both theoretical proof and empirical validation for the convergence of GCN training with both stale feature and stale feature gradients.
>
> Furthermore, we would like to point out that the proposed **smoothing method** (Section 4.3) is also **technically novel** and has not been considered by any existing methods (e.g., Dorylus [2]). We are the first to propose, develop, and validate such a method for effectively reducing the error caused by stale features or/and stale feature gradients.
>
> > Experimental comparisons with Dorylus may be included to show the benefits of using stale gradients.
>
> Following your suggestion, we compare PipeGCN with Dorylus [2] by training their adopted model (2-layer GCN, 128 hidden units) with their hyperparameters (120 epochs, 0.01 learning rate) on their adopted dataset Reddit, and provide the results below:
>
> |                   | Accuracy | Epoch Time |
> | ----------------- | -------- | ---------- |
> | Dorylus* (2 GPUs) | 95.13%   | 8.1s       |
> | PipeGCN (2 GPUs)  | 95.14%   | 5.0s       |
> | Dorylus* (4 GPUs) | 95.18%   | 8.7s       |
> | PipeGCN (4 GPUs)  | 95.16%   | 5.5s       |
>
> From the above table, we can observe that PipeGCN offers a similar accuracy as Dorylus while leading to a better efficiency. In addition, we would like to humbly remind you that comparison with Dorylus is not mandatory according to the policy of ICLR2022, as it is a contemporaneous work, i.e., published in OSDI this July. Finally, *Dorylus does not share the exact implementation details in their original paper, and only works for dedicated lambda GPU servers but not for regular GPU servers, whereas we don’t have access to lambda servers. Therefore, we implement the Dorylus design for regular servers to obtain the above results.
>
> > **Figure 1 and Figure 2 can be improved.** In Figure 1(b), the “…” for Part 2 should be removed and all communication tasks should be explicitly listed (e.g., communicate features, communicate gradients, and note that communicating gradients happens after some computation). In Figure 1(b) and (c), the computation and communication tasks should have iteration counts such that the pipelining idea can be understood. In Figure 2, “communicate for Next L1 backward” happens before the L1 backward of the current iteration, how is that possible? Is it a typo?
>
> Thanks again for your careful reading and sorry for the confusion. We will make it more clear in our revision.
>
> We would like to clarify that Figure 1(b) is a high-level abstraction and illustrates only coarse grains of ‘Communicate’ and ‘Compute’ for easy understanding and fast entry into our work. Thus, we did not list all detailed communication tasks here but deferred the readers to Figure 2 for more details.
>
> For Figure 1(b) and (c), the iteration counts have already been included and been shown right below the ‘Timeline’ bar.
>
> For Figure 2, it is correct. As shown in ‘Timeline’, we use ‘Previous Iteration’ and ‘Current Iteration’ to separate the ‘Communicate for Next L1 backward’ and ‘L1 Backward’, which is to show that the communication comes from previous iteration and serves for its next iteration (i.e., the next iteration of the previous iteration is current iteration) and thus ‘L1 Backward’ of current iterations happens later. To avoid such misunderstanding, we will change the color of ‘update’ in Figure 2 as it requires communication.

---

> > ### Author Response · Authors · 2021-11-21
> > **Response to Reviewer cPnX: Part 2**
> >
> >
> >
> > > **The influence of the smoothing technique can be better analyzed.** In Table 4, there is not a consistent winner among PipeGCN, PipeGCN-G and PipeGCN-F in model accuracy.
> >
> > Thanks for your comment. The motivation of the proposed smoothing method is to improve the convergence speed. The spirit of Table 4 is to demonstrate that PipeGCN and its variants do not hurt accuries. As such, we do not expect to find the best variant from Table 4 but compare them in terms of convergence speed (Figure 4). We will clarify and discuss this aspect in our revision.
> >
> > [1] Li, Youjie, et al. "Pipe-SGD: A decentralized pipelined SGD framework for distributed deep net training." arXiv preprint arXiv:1811.03619 (2018).
> >
> > [2] Thorpe, John, et al. "Dorylus: Affordable, Scalable, and Accurate GNN Training over Billion-Edge Graphs." arXiv preprint arXiv:2105.11118 (2021).

---

### Official Review · Reviewer_KmNe · 2021-11-02

**Correctness:** 3
**Technical Novelty And Significance:** 3
**Empirical Novelty And Significance:** 3
**Recommendation:** 6
**Confidence:** 3

**Details Of Ethics Concerns:**

No concern.

**Main Review:**

This paper proposes a distributed GCN training on large graphs named PipeGCN. Specifically, the authors hide the communication time by parallelizing the communication and computation process in each layer and using the stale information for parameter updates.  The idea seemed simple and straightforward, and experiments show that the algorithm can achieve up to 2.2x speedup. The authors provide the convergence proof for PipeGCN and propose two smoothing methods for faster convergence.

 Several limitations of this paper are listed as follows:
1. The proof of convergence should be justified better. The author didn’t claim how can the proposed model satisfy Assumptions 3.1-3.3, such as whether the loss function they chose satisfies the Lipschitz continuous condition mentioned in Assumption 3.1. In addition, the convergence proof is not applicable if PipeGCN uses the most commonly used ReLU activation function, as ReLU doesn’t satisfy Assumption 3.2.
2. Some statements in the paper are not clear enough.  For example, in Table 1 the authors didn’t mention AllReduce of Weight Gradient PipeGCN, whereas it appears in line 32 in Algorithm 1.


**Summary Of The Paper:**

This paper proposes a distributed full-graph GCN training method to speed up GCN training for large-scale graphs.  Experiments demonstrate its performance and efficiency. Convergence proof is also provided.

**Summary Of The Review:**

The authors designed a pipelined distributed GCN training algorithm to speed up large-scale GCN training and demonstrated the performance and efficiency with abundant experimental results. The paper can be accepted if the authors improve the convergence proof and modify the uncleared statements.

---

> ### Author Response · Authors · 2021-11-21
> **Response to Reviewer KmNe**
>
> > The paper can be accepted if the authors improve the convergence proof and modify the uncleared statements.
>
> Thank you for your detailed review and constructive feedback!
>
> > **The proof of convergence should be justified better.** The author didn’t claim how can the proposed model satisfy Assumptions 3.1-3.3, such as whether the loss function they chose satisfies the Lipschitz continuous condition mentioned in Assumption 3.1. In addition, the convergence proof is not applicable if PipeGCN uses the most commonly used ReLU activation function, as ReLU doesn’t satisfy Assumption 3.2.
>
> Following recent works (e.g., VR-GCN [1] and MVS-GCN [2]), we made **the same generally adopted assumptions 3.1-3.3** for convergence analysis while using ReLU as the activation function. As you suggested, we replace ReLU with **GELU [3] that satisfies Assumption 3.2**, and evaluate the convergence of PipeGCN on Reddit under the same setting as Table 3 of the paper. The convergence curves are as follows:
>
> | # of Epoch                       | 100    | 600    | 1100   | 1600   | 2100   | 2600   | 3000   |
> | -------------------------------- | ------ | ------ | ------ | ------ | ------ | ------ | ------ |
> | GCN with ReLu                    | 94.95% | 96.83% | 97.02% | 97.08% | 97.09% | 97.08% | 97.16% |
> | GCN with **GELU**                | 95.22% | 96.74% | 97.01% | 97.08% | 97.10% | 97.12% | 97.13% |
> | PipeGCN with ReLU ( 2  part.)    | 91.74% | 96.56% | 96.87% | 96.91% | 97.00% | 96.97% | 96.94% |
> | PipeGCN with **GELU** ( 2 part.) | 94.45% | 96.56% | 96.86% | 96.91% | 96.98% | 97.10% | 97.14% |
> | PipeGCN with ReLU (4  part.)     | 92.24% | 96.41% | 96.65% | 96.77% | 96.98% | 97.06% | 97.01% |
> | PipeGCN with **GELU** (4 part.)  | 94.22% | 96.51% | 96.77% | 96.94% | 97.07% | 97.02% | 96.98% |
>
> From the above table, we can see that the convergence with GELU is consistent with our theory and experiments in the paper.
>
> > Some statements in the paper are not clear enough. For Example, **In Table 1 the authors didn’t mention AllReduce of Weight Gradient PipeGCN, whereas it appears in line 32 in Algorithm 1.**
>
> Thanks again for your careful review. In Table 1, we compare the **Difference** between Pipe-SGD [4] and PipeGCN, where the former ‘Reduce Overhead’ of AllReduce of Weight Gradient whereas the latter ‘Reduce Overhead’ of Aggregation of Feature/Feature Gradient, i.e., PipeGCN **does NOT reduce overhead of AllReduce** of Weight Gradient, and thus should not be put in Table 1.
>
> Line 32 in Algorithm 1 does include this **AllReduce** of Weight Gradient, but it is **for completeness** of the training algorithm, where this AllReduce is orthogonal to PipeGCN’s focus (feature and feature gradients) and remains untouched.
>
> [1] Chen, Jianfei, Jun Zhu, and Le Song. "Stochastic training of graph convolutional networks with variance reduction." arXiv preprint arXiv:1710.10568 (2017).
>
> [2] Cong, Weilin, et al. "Minimal variance sampling with provable guarantees for fast training of graph neural networks." Proceedings of the 26th ACM SIGKDD International Conference on Knowledge Discovery & Data Mining. 2020.
>
> [3] Hendrycks, Dan, and Kevin Gimpel. "Gaussian error linear units (gelus)." arXiv preprint arXiv:1606.08415 (2016).
>
> [4] Li, Youjie, et al. "Pipe-SGD: A decentralized pipelined SGD framework for distributed deep net training." arXiv preprint arXiv:1811.03619 (2018).

---

### Decision · Program_Chairs · 2022-01-20

**Decision:**

Accept (Poster)

**Comment:**

The paper proposes PipeGCN, a system that uses pipeline parallelism to accelerate distributed training of large-scale graph convolutional neural networks. Like some pipeline-parallel methods (but unlike others), PipeGCN involves asynchrony in the sense that its features and feature-gradients can be stale. The paper provides theoretical guarantees on the convergence of PipeGCN in the presence of this staleness, which is a nice contribution in itself. In discussion, the reviewers found the work to be well-executed and sound. All reviewers recommended acceptance, and I concur with this consensus.